# EVALTREE: Profiling Language Model Weaknesses via Hierarchical Capability Trees

**Zhiyuan Zeng**[♡]   **Yizhong Wang**[♡♠]   **Hannaneh Hajishirzi**[♡♠]   **Pang Wei Koh**[♡♠]
[♡]Paul G. Allen School of Computer Science & Engineering, University of Washington
[♠]Allen Institute for Artificial Intelligence   zyzeng@cs.washington.edu

## Abstract

An ideal model evaluation should achieve two goals: identifying where the model fails and providing actionable improvement guidance. Toward these goals for language model (LM) evaluations, we formulate the problem of generating a *weakness profile*, a set of weaknesses expressed in natural language, given an LM's performance on every individual instance in a benchmark. We introduce a suite of quantitative assessments to compare different weakness profiling methods. We also introduce a weakness profiling method EVALTREE. EVALTREE constructs a capability tree where each node represents a capability described in natural language and is linked to a subset of benchmark instances that specifically evaluate this capability; it then extracts nodes where the LM performs poorly to generate a weakness profile. On the MATH and WildChat benchmarks, we show that EVALTREE outperforms baseline weakness profiling methods by identifying weaknesses more precisely and comprehensively. Weakness profiling further enables weakness-guided data collection, and training data collection guided by EVALTREE-identified weaknesses improves LM performance more than other data collection strategies. We also show how EVALTREE exposes flaws in Chatbot Arena's human-voter-based evaluation practice. To facilitate future work, we provide an interface that allows practitioners to interactively explore the capability trees built by EVALTREE.

    ⬡  **Code and Data**   github.com/Zhiyuan-Zeng/EvalTree
    🌐  **Web Interface**   zhiyuan-zeng.github.io/EvalTree

## 1 Introduction

An ideal model evaluation ought to achieve the goals of (1) identifying where the evaluated model fails in a human-interpretable way, and (2) providing actionable guidance to improve the model (Liang et al., 2023; Holtzman et al., 2023; Gu et al., 2024; Saxon et al., 2024). However, current model evaluations commonly treat diverse instances in a benchmark uniformly, reducing model performance to a single aggregate metric or coarse-grained, category-level metrics at best (Raunak et al., 2022). Doing so obscures the reality that a benchmark is heterogeneous, which evaluates diverse capabilities at varying granularities, and that model performance can vary significantly across these capabilities. For example, on the MATH benchmark (Hendrycks et al., 2021b), GPT-4o mini (OpenAI, 2024a) achieves an accuracy of 75.1% when calculating combinations and arrangements of elements, but only 49.1% when analyzing geometric relationships using trigonometric principles, as shown in Figure 1(a). As a result, current model evaluations often fail to achieve the two goals.

Inspired by the preceding observation, we formulate the problem of generating a *weakness profile*, a set of natural language descriptions of a model's weaknesses, given the model's performance on every individual benchmark instance. We focus on profiling language model (LM) weaknesses (Figure 1(a)). A weakness (e.g., *"analyzing geometric relationships using trigonometric principles"*) is a capability where the LM performs poorly on instances that test for this capability. Weakness profiles advance both goals of model evaluation: (1) they provide practitioners with an intuitive takeaway to interpret where an LM fails, based on

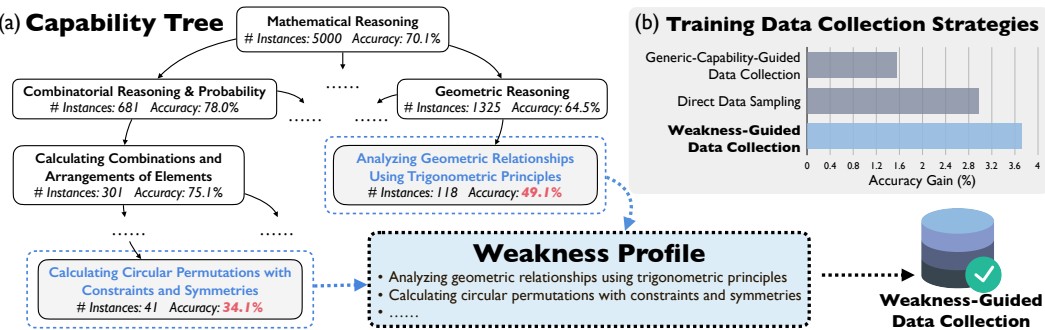

Figure 1: (a) EVALTREE automatically constructs a capability tree given an LM's performance on every individual benchmark instance, and then generates a weakness profile by extracting tree nodes with statistically low performance. (b) Training data collection guided by weakness profiling effectively improves LM performance, e.g., achieving an accuracy gain that is 2.5× larger than that obtained when being guided by a generic capability.

its heterogeneous performance across diverse capabilities; and (2) they are actionable, e.g., model developers can collect targeted training data to address the identified weaknesses.

In terms of how to profile LM weaknesses, manually analyzing LM performance on all instances is becoming increasingly unrealistic. Some works thus attempt to automatically profile LM weaknesses by constructing a single-level capability categorization across all benchmark instances and identifying low-performing categories (Murahari et al., 2024; Moayeri et al., 2024); however, fixed-granularity categorizations could be either too broad to provide precise diagnoses or too specific to retain high-level interpretability. More critically, while some methods, including those mentioned above, have been qualitatively shown to identify LM weaknesses, there is no existing study to compare them quantitatively.

To overcome these challenges, we establish a standard for what an ideal weakness profile should achieve and introduce **a suite of quantitative assessments**. We then propose **EVALTREE**, a weakness profiling method that automatically constructs a hierarchical tree for any LM benchmark, where each node represents a capability described in natural language and is linked to a subset of instances that specifically evaluate this capability. Instances linked to each node are partitioned into subsets corresponding to children's capabilities, which are further subdivided into more specific, finer-grained sub-capabilities at successive levels of the children's subtrees. EVALTREE then evaluates an LM's performance at every tree node, providing a *capability tree*. To generate a weakness profile, EVALTREE extracts tree nodes with statistically low performance and takes their capability descriptions (Figure 1(a)).

Our experiments show that EVALTREE advances both evaluation goals via weakness profiling: (1) EVALTREE profiles LM weaknesses more precisely and comprehensively than existing methods on the MATH and WildChat (Zhao et al., 2024a) benchmarks; (2) synthetic data generation guided by EVALTREE-identified weaknesses effectively improves LM performance, e.g., achieving an accuracy gain that is 2.5× larger than that obtained when being guided by a generic capability (Figure 1(b)). Furthermore, we show how EVALTREE uncovers abnormal LM rankings in Chatbot Arena, exposing flaws in its human-voter-based evaluation practice. We also provide an interface that lets practitioners interactively explore capability trees to facilitate future work. Finally, we discuss future directions, including improving capability trees and leveraging capability trees for potential applications.

## 1.1 Related Work

**Structured Categorization.** Structured categorization of benchmark instances is the essential idea behind EVALTREE. Murahari et al. (2024); Moayeri et al. (2024) automatically categorize benchmark instances into capability groups, providing single-level capability categorization structures. A small number of datasets are released with hierarchical structures defined by

their creators. For example, some provide shallow trees, e.g., a two-layer taxonomy (Wang et al., 2022; Bai et al., 2024; Zhong et al., 2024a); some adopt existing trees to guide data collection, such as ImageNet (Deng et al., 2009) using WordNet (Miller, 1994) and iNat2017 (Horn et al., 2018) using a biological taxonomy. Most related to our work, Wang et al. (2023); Zhong et al. (2024b) recursively cluster instances in a dataset to construct trees, and Anthropic's internal system Clio (Tamkin et al., 2024) employs Claude 3.5 (Anthropic, 2024) to build trees of human-LM conversations based on specific attributes or characteristics (e.g., topic). However, these techniques either incur prohibitively high LM usage costs or do not release key implementation details and source code, making them difficult to use.

**Automatic Weakness Identification.** Manually analyzing LM performance on instances in a benchmark for weakness profiling is becoming increasingly unrealistic. This is because LM benchmarks are growing in complexity to match the expanding versatility of emerging LMs; moreover, some datasets (e.g., WildChat (Zhao et al., 2024a)) collect real-world human-LM interactions, leading to the emergence of capabilities (tested within the benchmark) that are not foreseeable even by their creators in advance, further complicating manual efforts. Some works thus attempt to automatically profile LM weaknesses by using LMs to analyze evaluation results (Zhong et al., 2022) or by identifying low-performing categories from a single-level capability categorization (Murahari et al., 2024; Moayeri et al., 2024). Among these works, we are the first to formulate the problem of weakness profiling with quantitative assessments. Targeting similar goals, some works identify interpretable weaknesses (Eyuboglu et al., 2022; Hua et al., 2023), but assume closed output spaces, making them unsuitable for open-ended tasks; others (Wu et al., 2019; Ribeiro et al., 2020) propose interactive tools based on predefined failure modes, whereas we aim for fully automated profiling without such assumptions. Separately, while weakness profiling operates entirely on existing benchmarks and emphasizes interpretability, some prior work explores identifying model weaknesses by constructing custom instance sets to highlight underperforming areas (Ribeiro & Lundberg, 2022; Gao et al., 2023; Li et al., 2024; Wang et al., 2025).

## 2 LM Weakness Profiles

### 2.1 Definition and Desiderata

The problem of identifying LM weaknesses is broad. In this paper, we define a *weakness profile* in the simplest way that aligns with the two goals of identifying where an LM fails and providing improvement guidance. We let $\mathcal{C}$ denote the set of all possible natural language descriptions and assume an underlying data distribution $\mathcal{D}$. A weakness profile for an LM on a given benchmark drawn from the distribution $\mathcal{D}$ is a set $W = \{w_1, w_2, \ldots, w_M\} \subset \mathcal{C}$, where $M$ can vary among different profiles, and each *identified weakness* $w_i \in W$ is a natural language description of a capability, such as *"analyzing geometric relationships using trigonometric principles."* An ideal weakness profile $W$ satisfies three (informal) desiderata:

1. **Low-performance identification** (precision): The LM should exhibit low performance on instances (sampled from $\mathcal{D}$) testing for each identified weakness $w_i \in W$.
2. **Comprehensive coverage** (comprehensiveness): $W$ should reflect weaknesses that can be captured from the LM's performance on $\mathcal{D}$ as comprehensively as possible.
3. **Appropriate granularity**: Each $w_i$ should avoid being overly specific or generic.

We introduce concrete assessments in the next subsection to quantitatively compare weakness profiles along these desiderata and introduce experimental details in Section 5.

A weakness profiling method takes as input an LM's evaluation result on a given benchmark of size $N$ sampled from the data distribution $\mathcal{D}$, represented as a vector $g \in \mathbb{R}^N$, where each $g_i$ denotes the performance metric achieved by the LM on the $i$-th instance. We refer to this instance set as the *profiling set*. Since "weakness" is inherently a relative concept, a weakness profiling method should also include a user-tunable hyperparameter $\tau$ to control strictness; for example, increasing $\tau$ makes weakness identification less strict, allowing capabilities with relatively higher performance to be identified, whereas decreasing $\tau$ makes it more strict, restricting identification to the LM's most severe failures. When referring to a specific method in context, we denote $W_\tau$ as the weakness profile generated with a given $\tau$.

## 2.2 Assessment for Comparing Weakness Profiles

We assume the existence of a *test set* sampled from the data distribution $\mathcal{D}$. Furthermore, given a capability description $c \in \mathcal{C}$, we call an instance that tests for this capability an *associated instance* of $c$, with the index set of all associated instances in the test set denoted as $A(c)$. In our experiments, we prompt an LM to determine whether a given instance is an associated instance of a capability $c$ to get $A(c)$, with further details in Appendix E.1.

We introduce two assessments below to measure the effectiveness of a weakness profile in the first evaluation goal of identifying where an LM fails, based on the three desiderata.

**Low-Performance Identification Assessment.** We denote the LM's evaluation result vector on the test set as $f$, analogous to $g$ defined above for the profiling set. We also define the LM's performance metric over a set of instance indices $S$ as $F(S) = \sum_{x \in S} f_x / |S|$, assuming that the performance metric can be averaged; for example, each $f_i$ might be a binary value (0/1) indicating whether the LM correctly solved the $i$-th instance, in which case $F(S)$ is the accuracy of the LM on the set $S$. To measure desideratum 1, i.e., low-performance identification, we examine how low the average performance across identified weaknesses can be, computed as $\sum_{w_i \in W} F(A(w_i)) / |W|$. Denoting $S = \bigcup_{w_i \in W} A(w_i)$, we also compare how low $F(S)$ can be, i.e., the performance metric on all instances that test for at least one identified weakness in $W$. In the two comparisons, **a lower metric value indicates weaker performance on the identified weaknesses**, which can better satisfy desideratum 1.

**Ground-Truth Weakness Assessment.** To measure all three desiderata, inspired by Zhong et al. (2023), we generate a synthetic evaluation result for a "hypothetical" LM's performance on the profiling set. We use synthetic evaluation results rather than evaluation results of real LMs because desideratum 2, i.e., comprehensive coverage, cannot be reliably measured without prior knowledge of the LM's true weaknesses, which is exactly the problem we are trying to solve. By generating a synthetic evaluation result, we can control the *ground-truth* weaknesses and thus have such prior knowledge, allowing for a rigorous assessment. We start with a predefined ground-truth weakness profile $W^* = \{w_1^*, w_2^*, \ldots, w_{M^*}^*\}$. Then, we independently sample each $g_i$ such that instances associated with weaknesses in $W^*$ have systematically lower values of $g_i$ than others. Finally, **to assess a weakness profile $W$, we measure its alignment with the ground-truth profile $W^*$** based on the overlap of associated instances in the test set; we restrict $|W|$ to values that are not significantly larger than $|W^*|$, preventing methods from inflating scores by generating overly specific descriptions that increase $|W|$, which would violate desideratum 3, i.e., appropriate granularity.

**Extrinsic Assessment: Weakness-Guided Training Data Collection.** We examine the effectiveness of a weakness profile in supporting the second evaluation goal of improving the evaluated LM. In the real world, LM developers collect additional training data and perform finetuning to further improve an LM. A common strategy is to collect data guided by a generic capability such as "mathematical reasoning". We hypothesize that a *weakness-guided strategy*, wherein a weakness profile for the LM serves as actionable guidance for targeted data collection, may be more effective by directly addressing where the LM fails. For a controlled comparison, we collect data by synthetic data generation and **compare LMs trained on data generated under the guidance of different weakness profiles**.

# 3 EVALTREE: A Tree-Based Method for Profiling LM Weaknesses

## 3.1 Automatic Construction of Capability Trees

EVALTREE constructs a *capability tree* automatically. EVALTREE first constructs a tree that hierarchically organizes and interprets the capabilities tested within a benchmark. Each tree node represents a specific capability expressed in natural language and is linked to a subset of benchmark instances that evaluate this capability. The root node is linked to all instances, and each node's children together partition instances linked to it into subsets corresponding to more specific sub-capabilities, as shown in Figure 1(a). Finally, every leaf corresponds one-to-one with an individual instance; it is worth noting that instances linked to each node

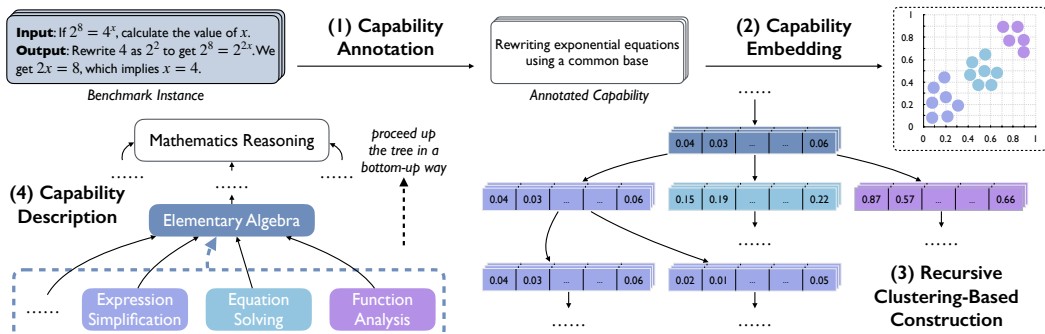

Figure 2: EVALTREE's four-stage tree construction pipeline. **(1) Capability Annotation** prompts an LM to identify a natural language description of each instance's capability. **(2) Capability Embedding** maps instances to a vector space using sentence embeddings of their annotated capabilities. **(3) Recursive Clustering-Based Construction** builds the tree by clustering capability embeddings using K-Means recursively. **(4) Capability Description** assigns each node a natural language summary of its children's capabilities using an LM.

are exactly the leaves in its subtree. We propose an automatic four-stage tree construction pipeline, which takes all instances of a benchmark as input, as shown in Figure 2.

**Stage (1) Capability Annotation** identifies the specific capability description required for each benchmark instance by prompting an LM, a practice also adopted in previous work analyzing LM capabilities (Ouyang et al., 2023; Didolkar et al., 2024; Kaur et al., 2024). The LM is asked to not mention the instance's specific content. See Figure 2 for an example.

**Stage (2) Capability Embedding** uses an off-the-shelf sentence embedding model to generate a capability embedding for each annotated capability from the stage (1).

**Stage (3) Recursive Clustering-Based Construction** recursively builds the hierarchical structure of the tree, starting from the root node linked to all instances. For each node, we cluster the capability embeddings of instances linked to it using K-Means (MacQueen, 1967). We iterate over cluster numbers from 2 to a predefined maximum value and select the one that yields the highest Silhouette score (Rousseeuw, 1987). This practice follows Katz et al. (2024), which also determines the cluster number automatically when the value is not predefined. Each cluster in the selected clustering becomes the set of instances linked to a newly created child node. The process continues recursively for each (non-leaf) child node.

**Stage (4) Capability Description** assigns a natural language description to each tree node to interpretably specify the capability represented by this node. For each leaf node (instance), we take its annotated capability directly as its capability description. For non-leaf nodes, we describe their capabilities at progressive granularities by proceeding up the tree in a bottom-up way, prompting an LM to summarize the capabilities of a node's children into a natural language description that captures their overarching scope; the LM's output is prompted to cover all children's capabilities without introducing extraneous concepts.

After constructing the tree, EVALTREE then provides a *capability tree* by evaluating LM performance at every node. Since each node is linked to a subset of benchmark instances, an evaluation practice can be seamlessly applied to this subset. For example, metrics such as accuracy or win-rate (Dubois et al., 2023) can be computed on instances linked to each node. See Appendix A and G for more details and an alternative tree construction approach.

## 3.2 Generating a Weakness Profile from the Capability Tree

EVALTREE generates an LM weakness profile by extracting nodes where the LM's performance metric is significantly below a user-tunable threshold $\tau$; for clarity, we consider the specific case of correctness-based accuracy being the metric. The extraction algorithm traverses the capability tree from the root to the leaves (see Appendix B for details):

1. **Statistical Test.** At each visited node, we perform a binomial test to determine whether its accuracy is significantly lower than $\tau$. The test uses the number of linked instances as the total sample size and the number of correctly solved instances as the count of successes. We apply the same test to the node's direct children[1].

2. **Node Extraction.** A visited node is extracted if: (a) it passes the test described above, **and** (b) all its direct children with sufficient instances (determined by a hyperparameter threshold of number) also pass the test. The design of (b) aims to identify the weakness at a granularity that is sufficiently specific. For example, if *"algebra"* performs statistically below the threshold overall but the LM performs well on its *"four-operations"* child while performing poorly on *"abstract algebra,"* identifying *"algebra"* as a weakness obscures the fact that the real weakness might lie in *"abstract algebra"* (or other sub-capabilities); here, further traversal is required.

3. **Stopping Criteria.** Traversal stops at a node if: (a) its instance number is smaller than a hyperparameter threshold, **or** (b) the node has been extracted.

Finally, the nodes extracted from running the algorithm are non-overlapping, i.e., no instance (leaf node) is linked to more than one extracted node. The final weakness profile consists of the capability descriptions of the extracted nodes. By adjusting the meaning of "count of successes" in the statistical test, this algorithm also supports various metrics (e.g., accuracy and win-rate) and can identify strengths (performance above a threshold).

# 4 Baseline Methods for Profiling LM Weaknesses

We describe the baseline methods, which are representative of existing methods that have been qualitatively shown to profile LM weaknesses. See Appendix D for additional details.

**TEXTDIFF** (Zhong et al., 2022) is an LM-based method that automatically describes differences between two text distributions in natural language. While not originally designed for weakness profiling, prior work has used it to describe distributional differences between two instance sets. We adapt this method by comparing instances where the evaluated LM fails versus succeeds, using the described differences to identify its weaknesses. Specifically, we randomly sample two sets of instances: those where the evaluation result indicates that the evaluated LM has failed, and those where it has succeeded. We then prompt a diagnostic LM using the sampled instances to output a predefined number of potential weaknesses that might cause the evaluated LM to struggle. We compute the evaluated LM's performance on the associated instances in the profiling set (Section 2.2) for each potential weakness and select those with the lowest performance metrics as the weakness profile. Note that this step actually **gives TEXTDIFF an unfair advantage** over other methods in our experiments, as it uses the identical implementation used by the method assessment to determine associated instances; however, a method should not have access to this information in principle, such as which LM is used or what prompt is used for method assessment.

**QUALEVAL** (Murahari et al., 2024) uses an automatic LM-based pipeline to derive a pre-defined number of capabilities (e.g., 20) described in natural language from all benchmark instances. The method then applies a linear programming algorithm to assign each benchmark instance to some of the derived capabilities. Finally, it outputs a single-level capability categorization structure. We compute the evaluated LM's performance metric on all instances (in the profiling set) assigned to each capability and identify a set of weaknesses as the weakness profile by selecting capabilities with the lowest performance metrics.

In these two methods, $\tau$ could be either the size of the weakness profile or a performance metric threshold, and the two can be transformed interchangeably.

---

[1]Note that setting a significance level of $\alpha$ for each node's statistical test does not guarantee an overall $1 - \alpha$ confidence level across all tests, as they are not corrected for multiple comparisons.

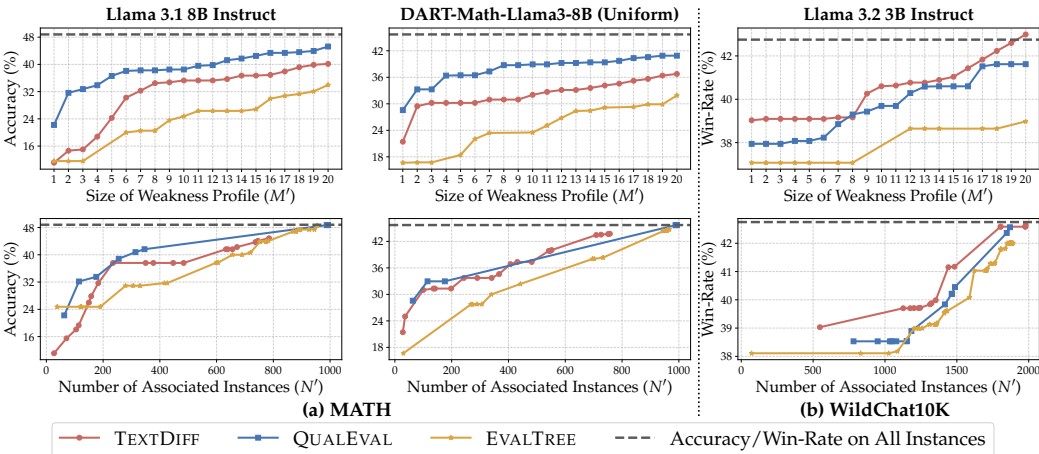

Figure 3: Comparison of weakness profiling methods using Low-Performance Identification Assessment. The first row shows how the average LM performance across identified weaknesses changes as we vary the minimum weakness profile size $M'$. The second row shows how the overall performance on all associated instances changes as we vary the minimum number of associated instances $N'$. Experiments in (a) were conducted on MATH with Llama 3.1 8B Instruct (Dubey et al., 2024) and DART-Math-Llama3-8B (Uniform) (Tong et al., 2024), and experiments in (b) were conducted on WildChat10K, where the win-rate is the percentage of instances in which Llama 3.2 3B Instruct (Meta, 2024) is preferred over Gemma 2 IT 2B (Rivière et al., 2024). A lower curve indicates more precise identification of true low-performing weaknesses and EVALTREE consistently achieves the lowest curve.

## 5 Experimental Results

We now present the results of our experiments that compare all weakness profiling methods, i.e., those introduced in Section 4 and EVALTREE, using the three assessments for weakness profiles introduced in Section 2.2. As preparation for the first two assessments, for each method, we sweep over $\tau$ to obtain a collection of all distinct weakness profiles $\{W_{\tau_1}, W_{\tau_2}, \ldots\}$, where each profile is included only once even if generated by multiple $\tau$.

### 5.1 Low-Performance Identification Assessment

Low-Performance Identification Assessment compares how low the LM's performance is on weaknesses identified by different methods. We assess all weakness profiling methods on the MATH (Hendrycks et al., 2021b) and WildChat10K (a subset we curated from WildChat (Zhao et al., 2024a)) benchmarks and randomly split each benchmark into profiling/test sets (see Appendix C for more configuration details). We constrain the minimum weakness profile size to compare the average performance across identified weaknesses and constrain the minimum number of associated instances to compare overall performance on all associated instances. To visualize the comparisons, we plot two curves in Figure 3: one with the minimum profile size $M'$ (ranging from 1 to 20) on the x-axis and $\min\{\sum_{w_i \in W_\tau} F(A(w_i))/|W_\tau| \mid \forall \tau, |W_\tau| \geq M'\}$ on the y-axis, and another with the minimum associated instance number $N'$ (ranging from 1 to the test set size) on the x-axis and $\min\{F(S_\tau) \mid \forall \tau, |S_\tau| \geq N'\}$ on the y-axis, where $S_\tau = \bigcup_{w_i \in W_\tau} A(w_i)$. EVALTREE consistently achieves the lowest curve, demonstrating its **superior precision in capturing true weaknesses** compared to other methods. See Appendix E.2 for qualitative analysis.

### 5.2 Ground-Truth Weakness Assessment

Ground-Truth Weakness Assessment compares how precisely and comprehensively different weakness profiling methods capture ground-truth weaknesses (on synthetic LM

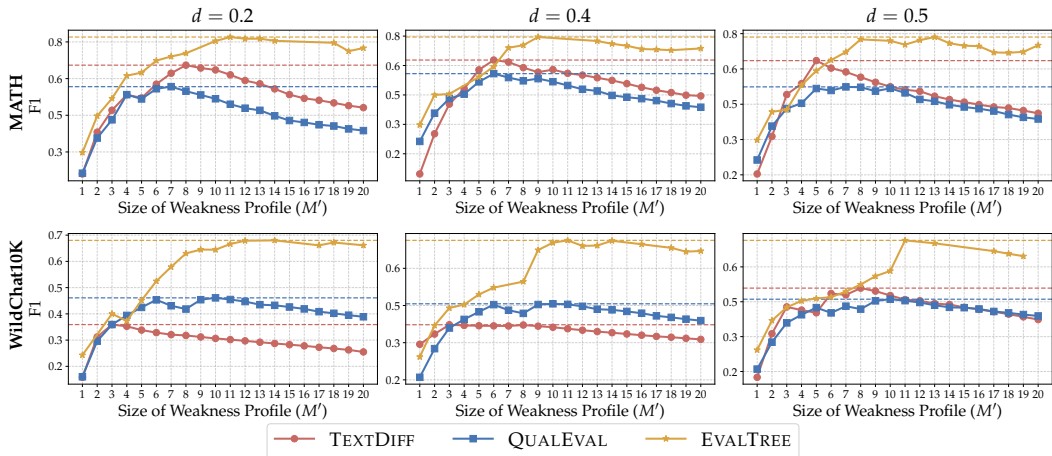

Figure 4: Comparison of weakness profiling methods using Ground-Truth Weakness Assessment. The plot shows F1 score curves of TEXTDIFF, QUALEVAL, and EVALTREE, where the weakness profile size varies from 1 to 20; the F1 score measures how precisely and comprehensively ground-truth weaknesses are captured. A horizontal line indicates each method's highest score. $d$ is a hyperparameter to control the sampling probability.

evaluation results) with appropriate description granularities. We manually curated 10 ground-truth weaknesses at various granularities for MATH and WildChat10K. For each benchmark, we generated three synthetic evaluation results by sampling with different hyperparameters that shape the probability distribution. For a given weakness profile, we compute the F1 score based on the overlap of associated instances to measure both precision and comprehensiveness relative to the ground-truth weakness profile $W^*$. We plot a curve with $M'$ (ranging from 1 to 20) on the x-axis and the F1 score of $W_\tau$, where $|W_\tau| = M'^2$, on the y-axis. All curves are shown in Figure 4 and Appendix E.3.3. We observe that **for most $M'$, the F1 scores achieved by EVALTREE surpass the highest F1 scores obtained by the other two methods**. For additional details and analysis, see Appendix E.3.1 and E.3.2.

## 5.3 Extrinsic Assessment: Weakness-Guided Training Data Collection

Extrinsic Assessment compares how effectively weakness profiles from different methods guide targeted training data collection to improve the evaluated LM; here, we conducted proof-of-concept experiments using a data-generation LM to generate (synthetic) data inputs (Kim et al., 2024) for data collection. The generic-capability-guided data collection strategy uses a description of the targeted benchmark's overall capability as guidance. For each weakness profiling method, we have a corresponding data collection strategy that randomly samples an identified weakness (in the weakness profile generated by the method) as guidance for generating each data input. For context, we also included the result in which training data inputs were directly sampled from the profiling set; however, we emphasize that this strategy has an inherently unfair advantage due to its distributional match to the test set and is **not a direct point of comparison** in our proof-of-concept experiments, which focus on LM developers' real-world practice of collecting new finetuning data.

We started with Llama 3.1 8B Instruct (Dubey et al., 2024) for MATH and DeepSeek-Coder-Base 6.7B (Guo et al., 2024) for DS-1000 (Lai et al., 2023), following configurations in Appendix C. When generating an input, we randomly sampled 5 inputs from the profiling set as in-context examples for the data-generation LM. We compared the performance of different LMs on the test set. For all data collection strategies, we collected the same amount of finetuning data inputs, with the output produced by separately feeding the input to

---

[2]If multiple thresholds $\tau$ for EVALTREE result in the same profile size, we select the lowest $\tau$. Note that the same profile size does not necessarily imply identical weakness profiles.

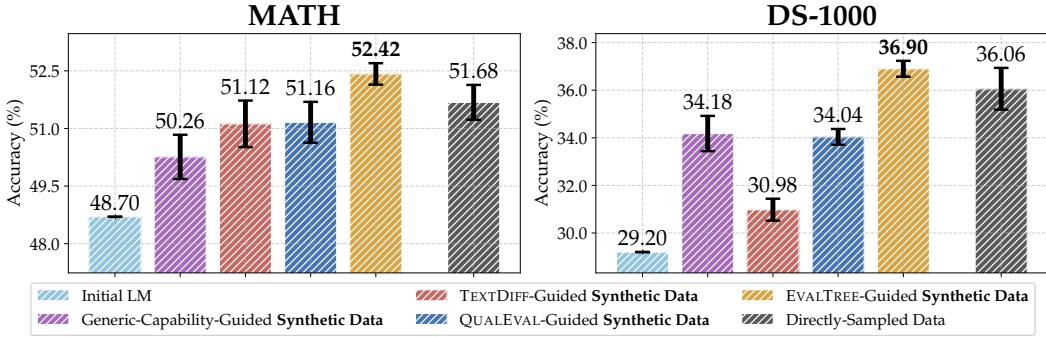

Figure 5: Accuracy of different LMs on MATH and DS-1000 test sets. Each chart includes the accuracy of the initial LM (Llama 3.1 8B Instruct and DeepSeek-Coder-Base 6.7B for MATH and DS-1000). For all other results, bars represent the accuracy of LMs trained on data collected by the corresponding strategy, with error bars indicating the standard error across 5 seeds. Bars for LMs trained on directly sampled data are included for reference, although they have an unfair advantage and are not a direct point of comparison. Data collection guided by EVALTREE-identified weaknesses yields the highest accuracy gain.

the data-generation LM. Refer to Appendix E.4 for more details. The results in Figure 5 demonstrate that the LM trained on EVALTREE-guided synthetic data significantly outperformed other LMs. Notably, the EVALTREE-guided data collection strategy even slightly outperformed directly sampling data from the profiling set. Therefore, **EVALTREE provides effective and targeted signals for guiding data collection to improve LM performance**.

### 5.4 LM Usage Cost Comparison

EVALTREE also incurs significantly lower LM usage costs than other methods. When each method identifies 20 weaknesses on MATH, the LM usage costs of TEXTDIFF and QUALEVAL were approximately 20 and 8 times higher than EVALTREE's cost, respectively. This occurs because EVALTREE's LM usage cost remains constant regardless of the weakness profile size $|W|$, whereas the costs of the others scale linearly with $|W|$. See Appendix E.5 for details.

### 5.5 Analysis on Threshold $\tau$ for EVALTREE's Node Extraction

We analyze how the choice of $\tau$ influences the nodes extracted by the algorithm in Section 3.2. We examine the LM performance on all extracted nodes as $\tau$ varies, referred to as *weakness/strength nodes*, i.e., nodes extracted by the algorithm where the LM's performance is significantly lower/higher than a given threshold $\tau$. To do this, we use the profiling set to build the capability tree and extract weakness/strength nodes with varying thresholds $\tau$. We locate the position of each instance in the test set on the capability tree by computing its capability embedding and then traversing from the root guided by the embedding. Specifically, at each non-leaf node, we predict the child cluster to which the instance belongs (by comparing its capability embedding with the K-Means clustering centers and then picking the closest one), determining which child's subtree to traverse into next; we call an instance that enters a weakness/strength node's subtree a *weakness/strength instance* and study LM performance on all weakness/strength instances from the test set as $\tau$ varies.

We experimented with the MATH, MMLU (Hendrycks et al., 2021a), DS-1000, and Wild-Chat10K benchmarks, and Figure 6, 7, 8, and 10(a) show the LMs' performance on weakness/strength instances. To further study generalizability, we experimented with two setups using different benchmarks as profiling and test sets; in the first setup, MATH is the profiling set and CollegeMath (Tang et al., 2024) is the test set; in the second setup, WildChat10K is the profiling set, and the test sets consisted of 10K instances we curated from ShareGPT, called ShareGPT10K, and a released subset of Chatbot Arena (Chiang et al., 2024), respectively; we show the results in Figure 9 and 10(b). See Appendix C for more configuration details. We

observe that LM performance on weakness/strength instances from the test set aligns well with the node extraction algorithm's goal. Specifically, performance on weakness/strength instances is generally below/above $\tau$. Furthermore, **as $\tau$ for extracting weakness/strength nodes decreases/increases, the performance on weakness/strength instances generally decreases/increases**, so $\tau$ is an effective hyperparameter for controlling strictness.

# 6 Further Applications of EVALTREE

Beyond identifying LM weaknesses, EVALTREE has broader applications in improving evaluation practices and facilitating LM capability analysis. We present two examples: (1) using EVALTREE to expose flaws in a widely used human-voter-based evaluation practice, and (2) implementing an interface for exploring capability trees to support future research.

**Identifying Flaws in Chatbot Arena Evaluation.** We give an application example by showing how EVALTREE exposes flaws in the human-voter-based evaluation practice of Chatbot Arena (Chiang et al., 2024). We begin by using EVALTREE to profile LM weaknesses on Chatbot Arena. To do this, we construct the capability tree for Chatbot Arena, where EVALTREE ranks 64 LMs at each node by computing Elo scores based on human comparison pairs for instances linked to the node; it then identifies weaknesses of strong LMs like GPT-4 (OpenAI, 2023) by extracting nodes where their ranking is unexpectedly low. The weakness profile reveals surprising patterns, leading us to discover that the identified weakness may not stem from the LM itself but from flaws in the evaluation practice. For instance, at the node *"Facilitating **inclusive, ethical, and strategic** communication and engagement across diverse and **sensitive** contexts,"* LMs such as Zephyr-7B-$\beta$ (Tunstall et al., 2023) and Alpaca 13B (Taori et al., 2023) rank significantly higher than GPT-4 and Claude 2.1 (Anthropic, 2023). We observed that this node contains many **user instructions with toxic requests**, where human voters tended to prefer models that provide toxic responses over well-aligned models that refuse to answer; more quantitative analysis is provided in Appendix F. This shows that the evaluation practice of Chatbot Arena allows uncontrolled user preferences to diverge from the values of LM development, producing potentially unreliable evaluation results. Because even minor misaligned preferences can significantly change LM rankings (Zhao et al., 2024b; Huang et al., 2025; Min et al., 2025), the need for improved evaluation practices is pressing. In this example, **EVALTREE provides actionable insights for refining evaluation practices**.

**User Interface of Capability Trees.** While the weakness profile provides a concise summary of where an LM fails, the full capability tree offers deeper and more comprehensive insights beyond this flat representation. Practitioners may wish to explore the capability tree itself to gain insights into a benchmark and analyze LM performance across capabilities at diverse granularities. To support this, we implement an interface that allows practitioners to interactively explore the capability trees constructed by EVALTREE. Users can expand a node to look deeper into its subtree, check the instances linked to the node, view its sub-capabilities represented by the node's children, examine LM performance at each node, etc. The interface provides an intuitive way for humans to navigate capability trees manually, establishing itself as a useful analysis tool. The interface is available **here**.

# 7 Future Work

Future work can enhance EVALTREE in several ways. For example, capability tree construction can be improved by optimizing the tree structure and capability descriptions, making its dimensionality and granularity more controllable by humans, exploring model-dependent hierarchical structures, and extending it beyond language to other modalities, etc. Additionally, it is useful to study how to quantitatively compare two capability trees directly. Beyond direct enhancements, capability trees can also support a variety of potential applications. For example, they can help analyze LM evaluation results to tailor benchmarks to specific needs, to provide actionable insights into training data mixture, etc. By moving beyond aggregate metrics from existing evaluations, EVALTREE enables a more comprehensive and interpretable analysis of LM performance across diverse capabilities, providing a useful foundation for future innovations in understanding and improving LM capabilities.

## Acknowledgments

We thank Zirui Cheng, Scott Geng, Joongwon Kim, Kyle Lo, Ian Magnusson, Sewon Min, Marco Tulio Ribeiro, Weijia Shi, Luca Soldaini, Ming Zhong, and Ruiqi Zhong for the insightful discussions. We thank Jacqueline He, Sandy Kaplan, Siting Li, Stella Li, Jiacheng Liu, Ben Newman, Rui Qiao, Rui Xin, and Lifan Yuan for proofreading the paper draft. We thank Hamish Ivison and Yuxuan Tong for sharing the model evaluation results. We thank members from the UW NLP and UW ML group for providing helpful feedback. We also thank All Hands AI's product OpenHands (Wang et al., 2024b) and Xingyao Wang for their help with web interface implementation. This work is supported by the Singapore National Research Foundation and the National AI Group in the Singapore Ministry of Digital Development and Information under the AI Visiting Professorship Programme (award number AIVP-2024-001); by the AI2050 program at Schmidt Sciences; by a Google ML and Systems Junior Faculty Award; by NSF Grant Nos. IIS2142739 and IIS2044660; by the Defense Advanced Research Projects Agency's (DARPA) SciFy program (Agreement No. HR00112520300); and by gift funding from Ai2.

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

## A  Implementation Details of Automatic Capability Tree Construction

This section provides additional details about the implementation of the automatic four-stage tree construction pipeline of EVALTREE, which is introduced in Section 3.1.

**Capability Annotation.** By default, we use OpenAI's `gpt-4o-mini-2024-07-18` (OpenAI, 2024a) in our experiments to generate natural language descriptions of the capabilities required to solve each benchmark instance. The prompt for the mathematics reasoning benchmarks (MATH (Hendrycks et al., 2021b) and CollegeMath (Tang et al., 2024)) is in Table 1; the prompt for MMLU (Hendrycks et al., 2021a) is in Table 2; the prompt for the Python code generation benchmark (DS-1000 (Lai et al., 2023)) is in Table 3; the prompt for the instruction-following benchmarks (WildChat10K (Zhao et al., 2024a), ShareGPT10K, and Chatbot Arena (Chiang et al., 2024)) is in Table 4. We set the max new tokens and temperature to 1024 and 0.0, respectively.

**Capability Embedding.** When generating capability embeddings, we prepend the prefix *"The model has the following skill or capability: "* to the annotated capability and feed the resulting sentence into a sentence embedding model. By default, we use OpenAI's `text-embedding-3-small` (OpenAI, 2024c) in our experiments.

**Recursive Clustering-Based Construction.** As we mentioned in the main text above, clusterings are generated for each cluster number from 2 to a predefined maximum value, and the Silhouette score[3] (Rousseeuw, 1987), which measures clustering quality based on cohesion and separation, is computed for each clustering. In our experiments, the predefined maximum value is set to 10 by default. One detail is that, if no clustering achieves a positive score, all instances linked to the current node are treated as leaves and become direct children of it. For the K-Means implementation, we use `sklearn.cluster.KMeans`[4].

**Capability Description.** By default, we use OpenAI's `gpt-4o-mini-2024-07-18` in our experiments to describe the specific capability each node represents in natural language. The prompt for the mathematics reasoning benchmarks (MATH and CollegeMath) is in Table 5;

---

[3]`https://scikit-learn.org/stable/modules/generated/sklearn.metrics.silhouette_score.html`. All hyperparameters are set to their default values.

[4]`https://scikit-learn.org/stable/modules/generated/sklearn.cluster.KMeans.html`. All hyperparameters are set to their default values.

**System Prompt**
Given a mathematical question and its correct solution, generate a gerund phrase that thoroughly and precisely describes the **specific** mathematical skill or capability required to solve the question.

- - - - - - - - - - - - - - - - - - - - - - - - - - - - - - - - - - - - - - -

**User Prompt**
## Question
{input}

## Solution
{output}

## Requirement
- The skill description should be an action-oriented gerund phrase that is **informative** and **detailed**.
- The phrase should refer to a **specific** skill or capability that comprehensively covers the key aspects of the solution, without including any context or specifics from the question or solution.
- Avoid unnecessary elements unrelated to the core capability.
- Please output **only a gerund phrase** describing the skill, with NO additional text.

Table 1:   The capability annotation prompt for the mathematics reasoning benchmarks (MATH (Hendrycks et al., 2021b) and CollegeMath (Tang et al., 2024)).

**System Prompt**
Given a multiple-choice question testing a model's wide-ranging knowledge and reasoning skills, generate a gerund phrase that thoroughly and precisely describes the **specific** skill or capability required to determine the correct answer.

- - - - - - - - - - - - - - - - - - - - - - - - - - - - - - - - - - - - - - -

**User Prompt**
## Question
{input}

## Answer
{output}

## Requirement
- The skill description should be an action-oriented gerund phrase that is **informative** and **detailed**.
- The phrase should refer to a **specific** skill or capability that comprehensively covers the key aspects of selecting the correct answer, without including any context or specifics from the question or answer.
- Avoid unnecessary elements unrelated to the core capability.
- Please output **only a gerund phrase** describing the skill, with NO additional text.

Table 2:   The capability annotation prompt for MMLU (Hendrycks et al., 2021a).

---

**System Prompt**
Given a code generation problem (involving data science) and its correct Python implementation, generate a gerund phrase that thoroughly and precisely describes the coding skill or capability required to solve the problem in detail.

- - - - - - - - - - - - - - - - - - - - - - - - - - - - - - - - - - - - - - - - - - - - - - - -

**User Prompt**
## Problem
{input}

## Implementation
{output}

## Requirement
- The skill description should be an action-oriented gerund phrase that is **informative** and **detailed**.
- The phrase should refer to a **specific** coding skill or capability that comprehensively covers the key aspects of the implementation, without including any context or specifics from the problem or implementation.
- Avoid unnecessary elements unrelated to the core capability.
- Please output **only a gerund phrase** describing the skill, with NO additional text.

---

Table 3: The capability annotation prompt for the Python code generation benchmark (DS-1000 (Lai et al., 2023)).

---

**System Prompt**
Given a user instruction and a reference response to the instruction, generate a gerund phrase that thoroughly and precisely describes the **specific** skill or capability required to respond to the instruction.

- - - - - - - - - - - - - - - - - - - - - - - - - - - - - - - - - - - - - - - - - - - - - - - -

**User Prompt**
## Instruction
{input}

## Response
{output}

## Requirement
- The skill description should be an action-oriented gerund phrase that is **informative** and **detailed**.
- The phrase should refer to a **specific** skill or capability that comprehensively covers the key aspects of the response, without including any context or specifics from the instruction or reference response.
- Avoid unnecessary elements unrelated to the core capability.
- Please output **only a gerund phrase** describing the skill, with NO additional text.

---

Table 4: The capability annotation prompt for the instruction-following benchmarks (WildChat10K (Zhao et al., 2024a), ShareGPT10K, and Chatbot Arena (Chiang et al., 2024)).

the prompt for MMLU is in Table 6; the prompt for the Python code generation benchmark (DS-1000) is in Table 7; the prompt for the instruction-following benchmarks (WildChat10K, ShareGPT10K, and Chatbot Arena) is in Table 8. We set the max new tokens and temperature to 1024 and 0.0, respectively.

---

**System Prompt**
Given a set of phrases, each summarizing the mathematical skills or capabilities needed to solve questions within a specific group, generate a gerund phrase that summarizes the collective set of mathematical skills or capabilities described across all groups.

- - - - - - - - - - - - - - - - - - - - - - - - - - - - - - - - - - - - - - - - - - - - - -

**User Prompt**
## Task
You are given a set of phrases, each summarizing the mathematical skills or capabilities needed to solve questions within a specific group. There are {group_number} groups in total. Your task is to **summarize** the collective set of mathematical skills or capabilities that represents the union of these descriptions in a detailed and informative manner.

## Skill Descriptions
{skill_descriptions}

## Requirements
- The output should be a **single gerund phrase** that succinctly summarizes the overarching mathematical skill or capability represented by the union of all the provided phrases.
- The output should comprehensively cover each skill description without going beyond them.
- The output should not simply enumerate the given phrases but instead provide a meaningful and informative summary of the mathematical skills or capabilities they collectively represent.
- Please output **only a gerund phrase** summarizing the mathematical skill or capability, with NO additional text.

---

Table 5: The capability description prompt for the mathematics reasoning benchmarks (MATH (Hendrycks et al., 2021b) and CollegeMath (Tang et al., 2024)).

## B Implementation Details of Extracting Nodes with Low Performance

Algorithm 1 provides the pseudocode for extracting nodes with significantly low accuracy on the capability tree (the algorithm introduced in Section 3.2). In the pseudocode, we use SIZE to indicate the number of instances linked to a node.

In our experiments, we use $\alpha = 0.05$, $\sigma_1 = 5$, and $\sigma_2 = 20$ by default.

This framework supports various metrics and deviation directions by adjusting the meaning of "total sample size" and "count of successes" in the statistical test step.

## C Default Experimental Configurations

This section provides the experimental configurations used in Section 5.

### C.1 Evaluation Results of LMs Across Different Benchmarks

For GPT-4o mini (OpenAI, 2024a) evaluation results on mathematics reasoning benchmarks, we run the generation ourselves; the system prompt is *"Please solve a math problem step-*

**System Prompt**
Given a set of phrases, each summarizing the skills or capabilities needed to answer multiple-choice questions testing broad knowledge and reasoning within a specific group, generate a gerund phrase that summarizes the collective set of skills or capabilities described across all groups.

**User Prompt**
## Task
You are given a set of phrases, each summarizing the skills or capabilities needed to answer multiple-choice questions testing broad knowledge and reasoning within a specific group. There are {group_number} groups in total. Your task is to **summarize** the collective set of skills or capabilities that represents the union of these descriptions in a detailed and informative manner.

## Skill Descriptions
{skill_descriptions}

## Requirements
- The output should be a **single gerund phrase** that succinctly summarizes the overarching skill or capability represented by the union of all the provided phrases.
- The output should comprehensively cover each skill description without going beyond them.
- The output should not simply enumerate the given phrases but instead provide a meaningful and informative summary of the skills or capabilities they collectively represent.
- Please output **only a gerund phrase** summarizing the skill or capability, with NO additional text.

Table 6: The capability description prompt for MMLU (Hendrycks et al., 2021a).

**System Prompt**
Given a set of phrases, each summarizing the coding skills or capabilities needed to solve code generation problems involving data science tasks within a specific group, generate a phrase that encapsulates the common coding skill or capability required across all the groups. The overall description should comprehensively cover each skill description without going beyond them, avoiding generic terms.

- - - - - - - - - - - - - - - - - - - - - - - - - - - - - - - - - - - - - - - - - - - - - - - - - - - - - - - - - -

**User Prompt**
## Task
You are given a set of phrases, each summarizing the coding skills or capabilities needed to solve code generation problems involving data science tasks within a specific group. There are {group_number} groups in total. Your task is to **summarize** the common coding skill or capability that represents the union of these descriptions in a detailed and informative manner.

## Skill Descriptions
{skill_descriptions}

## Requirements
The output should be a **single phrase** that succinctly summarizes the overarching coding skill or capability shared across all groups. It should not introduce any new concepts outside of those described in the provided phrases and must remain informative.

Please output **only a phrase** summarizing the skill or capability, with no additional text. Any output other than a phrase will NOT be accepted!

Table 7: The capability description prompt for the Python code generation benchmark (DS-1000 (Lai et al., 2023)).

---

**Algorithm 1** Extracting Nodes with Significantly Low Accuracy

---

**Input:** capability tree $T$, accuracy threshold $\tau$ {LM accuracy is pre-computed at each node of $T$ given the definition of a capability tree}
**Hyperparameter:** minimum node size $\sigma_1$ and $\sigma_2$, confidence level $\alpha$
**Output:** a set of extracted nodes $\mathcal{R}$
Initialize $\mathcal{R} \leftarrow \varnothing$
Initialize a map BINOMIALPASS $\leftarrow \{\}$ {Stores the binomial test result for each node}

{———————————— End of Initialization ————————————}
**First Pass: Binomial Test**
Define recursive function TESTNODE(*node*):
Perform a binomial test on *node* with accuracy threshold $\tau$ and confidence level $\alpha$
**if** the accuracy is significantly below $\tau$ at level $\alpha$ **then**
   BINOMIALPASS[*node*] $\leftarrow$ true
**else**
   BINOMIALPASS[*node*] $\leftarrow$ false
**end if**
**for** each *child* in *node.children* **do**
   TESTNODE(*child*)
**end for**
Call TESTNODE(*T.root*)

{———————————— End of First Pass ————————————}
**Second Pass: Node Extraction**
Define recursive function EXTRACTNODE(*node*):
**if** SIZE(*node*) $\geq \sigma_1$ **and** BINOMIALPASS[*node*] $=$ true **then**
   Initialize *allChildrenPass* $\leftarrow$ true
   **for** each *child* in *node.children* **do**
     **if** SIZE(*child*) $\geq \sigma_2$ **and** BINOMIALPASS[*child*] $=$ false **then**
       *allChildrenPass* $\leftarrow$ false
     **end if**
   **end for**
   **if** *allChildrenPass* $=$ true **then**
     Add *node* to $\mathcal{R}$
     Return {Skip its subtree to avoid overlap}
   **end if**
**end if**
**for** each *child* in *node.children* **do**
   EXTRACTNODE(*child*)
**end for**
Call EXTRACTNODE(*T.root*)
Output $\mathcal{R}$

{———————————— End of Second Pass ————————————}

---

---

**System Prompt**
Given a set of phrases, each summarizing the skills or capabilities needed to respond to instructions within a specific group, generate a gerund phrase that summarizes the collective set of skills or capabilities described across all groups.

- - - - - - - - - - - - - - - - - - - - - - - - - - - - - - - - - - - - - - - - -

**User Prompt**
## Task
You are given a set of phrases, each summarizing the skills or capabilities needed to respond to instructions within a specific group. There are {group_number} groups in total. Your task is to **summarize** the collective set of skills or capabilities that represents the union of these descriptions in a detailed and informative manner.

## Skill Descriptions
{skill_descriptions}

## Requirements
- The output should be a **single gerund phrase** that succinctly summarizes the overarching skill or capability represented by the union of all the provided phrases.
- The output should comprehensively cover each skill description without going beyond them.
- The output should not simply enumerate the given phrases but instead provide a meaningful and informative summary of the skills or capabilities they collectively represent.
- Please output **only a gerund phrase** summarizing the skill or capability, with NO additional text.

---

Table 8: The capability description prompt for the instruction-following benchmarks (WildChat10K (Zhao et al., 2024a), ShareGPT10K, and Chatbot Arena (Chiang et al., 2024)).

by-step. *Break down each step logically and reason through intermediate steps until reaching the final solution."*, and the user prompt is the question; we use `gpt-4o-mini-2024-07-18`, and set the max new tokens and temperature to 1024 and 0.0, respectively. For Llama 3.1 8B Instruct (Dubey et al., 2024) evaluation results, we also run the generation ourselves; we use the default system prompt, append the suffix *"Please reason step by step, and put your final answer within \\boxed{}."* to the question and set the max new tokens and temperature to 1024 and 0.0, respectively; the vLLM library (Kwon et al., 2023) is used to accelerate generation. Their generations are evaluated by our internal evaluation toolkit. We directly adopt DART-Math-Llama3-8B (Uniform) (Tong et al., 2024) evaluation results provided by the authors of its original paper.

For the evaluation results of all models on MMLU (Hendrycks et al., 2021a), we directly adopt the evaluation results provided by the authors of TÜLU 3 (Lambert et al., 2024).

MMLU (Hendrycks et al., 2021a) and CollegeMath (Tang et al., 2024) provide only the final answer to each question, but not the solution (reference output) needed for all weakness profiling methods. To address this, we take the response generated by GPT-4o mini as the reference output, which may have errors.

For DeepSeek-Coder-Base 6.7B (Guo et al., 2024) evaluation result on DS-1000 (Lai et al., 2023), we use the scripts provided by the DS-1000 GitHub repository [5] for generation, with vLLM added to accelerate generation. For GPT-4o (OpenAI, 2024b) and GPT-3.5 Turbo (OpenAI, 2022) evaluation results, we directly evaluate the generations of `gpt-4o-2024-08-06` and `gpt-3.5-turbo-0613` provided by the GitHub repository. In both cases, we use the scripts provided by the DS-1000 GitHub repository for evaluation.

---

[5] https://github.com/xlang-ai/DS-1000

To build the WildChat10K and ShareGPT10K benchmarks, we start with the publicly released versions of WildChat (Zhao et al., 2024a) and ShareGPT from HuggingFace Datasets[6] [7]; for both datasets, we keep only first-round conversations to collect instruction-response pairs, filter pairs where the combined length of the instruction and response exceeds 4096 Llama 3.2 tokens, and deduplicate the instructions; finally, we randomly sample 10K instruction-response pairs. For Chatbot Arena (Chiang et al., 2024), we use the publicly released version from HuggingFace Datasets[8]; for each instruction, we retain it only once and assign its reference output as the response from the strongest model (indicated by the overall ranking) for it; we finally have 44,230 instances in the Chatbot Arena benchmark.

In the instruction-following setup (Ouyang et al., 2022), where LMs respond to a set of free-form user instructions, the responses are commonly evaluated using the LM-as-a-judge paradigm (Zheng et al., 2023; Dubois et al., 2023), in which a significantly stronger LM serves as a judge by comparing responses produced by two LMs to the same instruction to determine which one is better. This produces a win-rate for each LM, ranging from 0% to 100%, representing the proportion of instances where its response is chosen as the better one. A higher win-rate is generally interpreted as a signal of better overall performance. When using the LM-as-a-judge paradigm, we use `gpt-4o-mini-2024-07-18` (OpenAI, 2024a) as the judge. The prompt for the LM judge is provided in Table 9, and we set the max new tokens and temperature to 50 and 0.0, respectively. Following Zeng et al. (2024), we compare each pair of responses to an instruction by querying the LM judge twice, swapping the order of the responses; this is due to potential positional bias (Wang et al., 2024a; Zeng et al., 2024), which can influence judgments based on the response order. For win-rate computation, we average the results of all comparisons. When using win-rate as the evaluation metric in the node extraction algorithm introduced in Section 3.2, the total sample size for the binomial test is twice the number of instances, and the count of successes corresponds to the number of times that one model's output is preferred or not preferred.

When running Llama 3.2 3B Instruct (Meta, 2024) and Gemma 2 IT 2B (Rivière et al., 2024) on instruction-following benchmarks (WildChat10K, ShareGPT10K, and Chatbot Arena), we use the default system prompt, directly use the instruction as the user prompt, and set the max new tokens and temperature to 4096 and 0.0, respectively. The vLLM library is also utilized to accelerate generation.

## C.2 Profiling/Test Splits

In Sections 5.1, 5.3, and 5.5, whenever the profiling and test sets originate from the same individual benchmark, we apply the following random profiling/test splits: the MATH benchmark was randomly partitioned into a 4000/1000 split, the MMLU benchmark into a 10042/4000 split, the DS-1000 benchmark into a 600/400 split, and the WildChat10K benchmark into an 8000/2000 split to create the profiling and test sets. In Section 5.5, the full sets of benchmarks are used in the cross-benchmark generalization setup.

# D  Implementation Details of Baseline Methods for Profiling LM Weaknesses

This section provides additional details about the implementation of baselines we assessed for profiling LM weaknesses, which are introduced in Section 4.

## D.1  Implementation Details of TEXTDIFF

When sampling instances where the evaluated LM has succeeded/failed, the sampling pool consists of all instances where the evaluated LM's correctness is correct/incorrect

---

[6]WildChat: `https://huggingface.co/datasets/allenai/WildChat`
[7]ShareGPT: `https://huggingface.co/datasets/anon8231489123/ShareGPT_Vicuna_unfiltered/blob/main/ShareGPT_V3_unfiltered_cleaned_split_no_imsorry.json`
[8]`https://huggingface.co/datasets/potsawee/chatbot-arena-llm-judges`

**System Prompt**
You are a helpful assistant in evaluating the quality of the outputs for a given instruction. Your goal is to select the best output for the given instruction.

---

**User Prompt**
Select the Output (a) or Output (b) that is better for the given instruction. The two outputs are generated by two different AI chatbots respectively.

Do NOT provide any explanation for your choice.
Do NOT say both / neither are good.
You should answer using ONLY "Output (a)" or "Output (b)". Do NOT output any other words.

# Instruction:
{instruction}

# Output (a):
{response_1}

# Output (b):
{response_2}

# Which is better, Output (a) or Output (b)? Your response should be either "Output (a)" or "Output (b)":

Table 9: The prompt for the LM judge.

for correctness-based accuracy, and for win-rate, all instances where the LM judge prefers the evaluated LM's response in both orders/does not prefer the evaluated LM's response in either order (before and after swapping the response order; see Appendix C). In our experiments, we sample 50 failed instances and 50 successful instances due to the context length limit. We then prompt GPT-4o (`gpt-4o-2024-08-06`) (OpenAI, 2024b) as the diagnostic LM using the sampled 50+50=100 instances. The prompts for MATH, WildChat10K, and DS-1000 are provided in Table 10, 11, and 12, respectively. We set the max new tokens and temperature to 4096 and 0.0, respectively. The diagnostic LM is asked to identify 20 (potential) weaknesses given these sampled instances. Then, we determine the associated instances (in the profiling set) for each outputted potential weakness, following the implementation described in Appendix E.1. We finally compute the performance metric on the associated instances for each potential weakness and identify a set of weaknesses as the weakness profile by selecting those with the lowest performance metrics.

### D.2 Implementation Details of QUALEVAL

As the authors of Murahari et al. (2024) have not released the code yet before we released this paper, we implemented QUALEVAL ourselves based on our scenario.

QUALEVAL starts with all instances in the benchmark, denoted as $\mathcal{B}$. All instances are first randomly partitioned into $\lceil \frac{|\mathcal{B}|}{k} \rceil$ chunks (we use $k = 20$ in all of our experiments), with each chunk size being no more than $k$, and each chunk is fed to `gpt-4o-mini-2024-07-18` (OpenAI, 2024a) to summarize a list of capabilities for instances in the chunk. The prompts used here for MATH, WildChat10K, and DS-1000 are provided in Table 13, 14 and 15, respectively. We set the max new tokens and temperature to 4096 and 0.0, respectively. We concatenate all capabilities generated for each chunk, getting a long list of capabilities for this benchmark.

We then iteratively shrink the list to get a final list of $m$ capabilities (we use $m = 20$ in our experiments). In each iteration, we split the list into multiple $mp$-size chunks (we use $p = 4$ in our experiments), and prompt `gpt-4o-mini-2024-07-18` to shrink each chunk into

**System Prompt**
Given a set of mathematics questions and their corresponding correct solutions, identify the specific weaknesses of a model.

You are provided with 50 mathematics questions that the model fails to solve and 50 mathematics questions that the model successfully solves. Based on this data, analyze and describe the model's weaknesses by identifying the high-level mathematical capabilities that the model struggles with. Group similar weaknesses under broader categories where applicable.

- - - - - - - - - - - - - - - - - - - - - - - - - - - - - - - - - - - - - - - - - - - - - - - -

**User Prompt**
## Task
You are given 50 mathematics questions that the model fails to solve and their corresponding correct solutions, along with 50 questions that the model successfully solves. Analyze and describe the weaknesses of the model by identifying specific high-level mathematical capabilities it struggles with, summarizing any related weaknesses under broader categories.

## Questions and Solutions
### Failed Cases
{negative_inputs_and_outputs}

### Successful Cases
{positive_inputs_and_outputs}

## Requirements
- **Output exactly 20 weaknesses.**
- Each weakness should be an **informative and detailed phrase** that refers to a **specific skill or capability** comprehensively covering key aspects of the failure, without including any specifics from the questions or solutions.
- Where possible, group related weaknesses under a single broader weakness category.
- Output each capability as a standalone phrase, with **no additional text, prefixes, symbols, or notations** on any line. For example, do NOT include numbered list markers, numerical prefixes, or numeric labels (e.g., '1.', '2.', etc.) in the output.

Table 10: The diagnostic LM prompt for MATH (Hendrycks et al., 2021b) used by TEXTDIFF.

**System Prompt**
Given a set of user instructions and their corresponding reference responses, identify the specific weaknesses of a model.

You are provided with 50 user instructions and their corresponding reference responses that the model fails to address effectively, and 50 user instructions and their corresponding reference responses that the model addresses successfully. Based on this data, analyze and describe the model's weaknesses by identifying the high-level capabilities it struggles with. Group similar weaknesses under broader categories where applicable.

- - - - - - - - - - - - - - - - - - - - - - - - - - - - - - - - - - - - - - - - - - -

**User Prompt**
## Task
You are given 50 user instructions and their corresponding reference responses that the model fails to address effectively, along with 50 user instructions and their corresponding reference responses that the model addresses successfully. Analyze and describe the weaknesses of the model by identifying specific high-level capabilities it struggles with, summarizing any related weaknesses under broader categories.

## User Instructions and Reference Responses
### Failed Cases
{negative_inputs_and_outputs}

### Successful Cases
{positive_inputs_and_outputs}

## Requirements
- **Output exactly 20 weaknesses.**
- Each weakness should be phrased as a specific capability, avoiding negative phrasing such as "lack," "difficulty," or similar terms.
- Each weakness should be an **informative and detailed phrase** that refers to a **specific skill or capability** comprehensively covering key aspects of the failure, without including any specifics from the instructions or reference responses.
- Where possible, group related weaknesses under a single broader weakness category.
- Output each capability as a standalone phrase, with **no additional text, prefixes, symbols, or notations** on any line. For example, do NOT include numbered list markers, numerical prefixes, or numeric labels (e.g., '1.', '2.', etc.) in the output.

Table 11: The diagnostic LM prompt for WildChat10K (Zhao et al., 2024a) used by TEXTDIFF.

**System Prompt**
Given a set of Python coding problems (involving data science) and their corresponding correct Python implementations, identify the specific weaknesses of a model.

You are provided with 50 code generation problems that the model fails to solve and 50 code generation problems that the model successfully solves. Based on this data, analyze and describe the model's weaknesses by identifying the high-level coding capabilities (related to data science) that the model struggles with. Group similar weaknesses under broader categories where applicable.

- - - - - - - - - - - - - - - - - - - - - - - - - - - - - - - - - - - - - - - - - - - -

**User Prompt**
## Task
You are given 50 Python coding problems (involving data science) that the model fails to solve and their corresponding correct Python implementations, along with 50 coding problems that the model successfully solves. Analyze and describe the weaknesses of the model by identifying specific high-level coding capabilities it struggles with, summarizing any related weaknesses under broader categories.

## Problems and Implementations
### Failed Cases
{negative_inputs_and_outputs}

### Successful Cases
{positive_inputs_and_outputs}

## Requirements
- **Output exactly 20 weaknesses.**
- Each weakness should be phrased as a specific capability, avoiding negative phrasing such as "lack," "difficulty," or similar terms.
- Each weakness should be an **informative and detailed phrase** that refers to a **specific skill or capability** comprehensively covering key aspects of the failure, without including any specifics from the code problems or implementations.
- Where possible, group related weaknesses under a single broader weakness category.
- Output each capability as a standalone phrase, with **no additional text, prefixes, symbols, or notations** on any line. For example, do NOT include numbered list markers, numerical prefixes, or numeric labels (e.g., '1.', '2.', etc.) in the output.

Table 12: The diagnostic LM prompt for DS-1000 (Lai et al., 2023) used by TEXTDIFF.

$m$ capabilities. The prompts used here for MATH, WildChat10K, and DS-1000 are provided in Table 16, 17 and 18, respectively. We set the max new tokens and temperature to 4096 and 0.0, respectively. After multiple iterations, this finally ends up with $m$ capabilities.

After deriving $m = 20$ capabilities in natural language from all benchmark instances, QUAL-EVAL assigns a relevance score to each pair of benchmark instances and capabilities, indicating the relevance of the instance to the capability. The score is an integer ranging from 1 to 5, where 5 indicates strong relevance and 1 indicates no relevance. This is done by prompting `gpt-4o-mini-2024-07-18` with each instance and the list of all derived capabilities, which outputs a list of scores for all instance-capability pairs for this instance. The prompts used here for MATH, WildChat10K, and DS-1000 are provided in Table 19, 20 and 21, respectively. We set the max new tokens and temperature to 4096 and 0.0, respectively.

After scoring each pair of benchmark instances and capabilities, QUALEVAL assigns each instance to exactly 2 capabilities to maximize the sum of the relevance scores of the chosen pairs (instance and assigned capability). The assignment is constrained such that the number of instances assigned to each capability is roughly proportional to the sum of its relevance scores across all instances. We use linear programming to perform the assignment, implemented with `scipy.optimize.linprog`[9]. Finally, QUALEVAL computes the performance metric for each capability, i.e., the performance metric on all its assigned instances, and identifies the capabilities with the lowest performance metrics as the weakness profile.

---

**System Prompt**
Given a set of mathematics questions and their corresponding correct solutions, identify the high-level mathematical capabilities required to solve these questions. Group similar capabilities where relevant.

- - - - - - - - - - - - - - - - - - - - - - - - - - - - - - - - - - - - - - - - -

**User Prompt**
## Task
You are given {instance_num} mathematics questions and their corresponding correct solutions. Identify the high-level mathematical capabilities required to solve these questions, summarizing any related capabilities under broader categories.

## Questions and Solutions
{inputs_and_outputs}

## Requirements
- Each capability should be an **informative and detailed phrase** that refers to a **specific skill or capability** comprehensively covering key aspects of the solution, without including any specifics from the questions or solutions.
- Where possible, group related capabilities under a single broader capability.
- Output each capability as a standalone phrase, with **no additional text, prefixes, symbols, or notations** on any line.

Table 13: The capability initialization prompt for MATH (Hendrycks et al., 2021b) used by QUALEVAL.

## E   Experimental Details of Assessing Weakness Profiling Methods

This section provides additional details about Section 5.

### E.1   Details of Determining Associated Instances

As described in Section 2.2, we prompt `gpt-4o-mini-2024-07-18` (OpenAI, 2024a) to determine whether an instance tests for a given capability (if yes, the instance is called an

---

[9]https://docs.scipy.org/doc/scipy/reference/generated/scipy.optimize.linprog.html. All hyperparameters are set to their default values.

**System Prompt**
Given a set of user instructions and their corresponding reference responses, identify the high-level capabilities required to respond effectively to these instructions. Group similar capabilities where relevant.

---

**User Prompt**
## Task
You are given {instance_num} user instructions and their corresponding reference responses. Identify the high-level capabilities required to respond effectively to these instructions, summarizing any related capabilities under broader categories.

## User Instructions and Reference Responses
{inputs_and_outputs}

## Requirements
- Each capability should be an **informative and detailed phrase** that refers to a **specific skill or capability** comprehensively covering key aspects of the response, without including any specifics from the instructions or reference responses.
- Where possible, group related capabilities under a single broader capability.
- Output each capability as a standalone phrase, with **no additional text, prefixes, symbols, or notations** on any line.

Table 14: The capability initialization prompt for WildChat10K (Zhao et al., 2024a) used by QUALEVAL.

**System Prompt**
Given a set of Python coding problems and their corresponding correct implementations, identify the high-level programming capabilities required to solve these problems. Group similar capabilities where relevant.

---

**User Prompt**
## Task
You are given {instance_num} Python coding problems and their corresponding correct implementations. Identify the high-level programming capabilities required to solve these problems, summarizing any related capabilities under broader categories.

## Problems and Implementations
{inputs_and_outputs}

## Requirements
- Each capability should be an **informative and detailed phrase** that refers to a **specific skill or capability** comprehensively covering key aspects of the solution, without including any specifics from the problems or implementations.
- Where possible, group related capabilities under a single broader capability.
- Output each capability as a standalone phrase, with **no additional text, prefixes, symbols, or notations** on any line.

Table 15: The capability initialization prompt for DS-1000 (Lai et al., 2023) used by QUALEVAL.

**System Prompt**
Given a list of mathematics capabilities, generate a shorter list of the most critically relevant capabilities by combining related items where appropriate.

---

**User Prompt**
## Task
You are given {current_num_capabilities} mathematics capabilities. Generate a list of no more than 20 capabilities by merging related capabilities into broader items where relevant.

## Capabilities
{capability_list}

## Requirements
- You should output **up to 20 capabilities**, ideally exactly 20.
- Each capability should be an **informative and concise phrase** that represents a **specific skill or capability** while covering key aspects of the capabilities provided.
- Consolidate related capabilities into a single, broader capability wherever possible to reduce the list length.
- Output each capability as a standalone phrase, with **no additional text, prefixes, symbols, or notations** on any line.

Table 16: The capability shrinking prompt for MATH (Hendrycks et al., 2021b) used by QUALEVAL.

**System Prompt**
Given a list of capabilities required for responding to user instructions, generate a shorter list of the most critically relevant capabilities by combining related items where appropriate.

---

**User Prompt**
## Task
You are given {current_num_capabilities} capabilities related to responding to user instructions. Generate a list of no more than 20 capabilities by merging related capabilities into broader items where relevant.

## Capabilities
{capability_list}

## Requirements
- You should output **up to 20 capabilities**, ideally exactly 20.
- Each capability should be an **informative and concise phrase** that represents a **specific skill or capability** while covering key aspects of the capabilities provided.
- Consolidate related capabilities into a single, broader capability wherever possible to reduce the list length.
- Output each capability as a standalone phrase, with **no additional text, prefixes, symbols, or notations** on any line.

Table 17: The capability shrinking prompt for WildChat10K (Zhao et al., 2024a) used by QUALEVAL.

---

**System Prompt**
Given a list of capabilities required for solving Python coding problems, generate a shorter list of the most critically relevant capabilities by combining related items where appropriate.

- - - - - - - - - - - - - - - - - - - - - - - - - - - - - - - - - - - - - - - - - - - - - -

**User Prompt**
## Task
You are given {current_num_capabilities} capabilities related to solving Python coding problems. Generate a list of no more than 20 capabilities by merging related capabilities into broader items where relevant.

## Capabilities
{capability_list}

## Requirements
- You should output **up to 20 capabilities**, ideally exactly 20.
- Each capability should be an **informative and concise phrase** that represents a **specific programming skill or capability** while covering key aspects of the capabilities provided.
- Consolidate related capabilities into a single, broader capability wherever possible to reduce the list length.
- Output each capability as a standalone phrase, with **no additional text, prefixes, symbols, or notations** on any line.

---

Table 18: The capability shrinking prompt for DS-1000 (Lai et al., 2023) used by QUALEVAL.

*associated instance*), which is a basic operation used in our assessments and TEXTDIFF. The prompts used here for MATH and WildChat10K are provided in Table 22 and Table 23, respectively; we also provide the prompt for DS-1000 in Table 24, used in experiments of Section 5.3. We set the max new tokens and temperature to 128 and 0.0, respectively.

## E.2 Qualitative Analysis of Low-Performance Identification Assessment

Table 25 presents the identified weaknesses from TEXTDIFF, QUALEVAL, and EVALTREE when the weakness profile size is 10, along with the LM performance on the associated instances (in the test set) of each identified weakness; they are based on applying the three methods to Llama 3.1 8B Instruct (Dubey et al., 2024) evaluation result on MATH (see Section 5.1). We observe that EVALTREE-identified weakness descriptions are generally more specific than those identified by the other two methods, enabling a more precise diagnosis and thus capturing capabilities where the LM exhibits lower performance.

## E.3 Experimental Details of Ground-Truth Weakness Assessment

### E.3.1 *Details of the Assessment Setup*

This subsection provides additional details about the setup of Ground-Truth Weakness Assessment in Section 5.2, based on the setup introduced in Section 2.2.

We used two benchmarks as testbeds, the MATH benchmark (Hendrycks et al., 2021b) and the WildChat10K benchmark (Zhao et al., 2024a). As described above, we manually curated a set of 10 ground-truth weaknesses (described in natural language) at diverse granularities as the ground-truth weakness profile, for MATH and WildChat10K, respectively. The ground-truth weakness profiles for MATH and WildChat10K are provided in Table 26 and Table 27, denoted as $W^*$. We aim to generate a synthetic evaluation result (on the profiling set) $g$ where the actual weaknesses are exactly this predefined ground-truth weakness profile $W^*$. First, we identify the associated instances for each ground-truth weakness. We then define two hyperparameters, the *base probability* $p \in (0, 1]$ and the *decrease rate* $d \in (0, 1)$,

**System Prompt**
Given a mathematics question with its solution and a numbered list of mathematical capabilities, rate each capability on a scale of 1-5 to indicate its relevance in solving this question. A score of 5 means the capability is very used, while 1 means it is not used at all.

- - - - - - - - - - - - - - - - - - - - - - - - - - - - - - - - - - - - - - - - - - - - - - -

**User Prompt**
## Task
You are given a mathematics question and solution, along with a list of 20 mathematical capabilities. For each capability, rate the degree to which it is required to solve this question.

## Question
{input}

## Solution
{output}

## Capabilities
{capability_list}

## Requirements
- For each capability, provide an integer **score from 1 to 5**. A score of 5 means the capability is very used, while 1 means it is not used at all.
- Include a brief **reasoning** for each score, explaining how you determined the score.
- Output the result in **JSON format** as follows:

```json
{
  "1": {"reasoning": "THE REASONING", "score": SCORE},
  "2": {"reasoning": "THE REASONING", "score": SCORE},
  "3": {"reasoning": "THE REASONING", "score": SCORE},
  ...
}
```

- Do NOT include any additional text outside of the JSON format, as **I will directly use 'json.loads' in Python to convert your output to a dictionary object**.

Table 19: The scoring prompt for MATH (Hendrycks et al., 2021b) used by QUALEVAL.

**System Prompt**
Given a user instruction with its reference response and a numbered list of capabilities, rate each capability on a scale of 1-5 to indicate its relevance in responding to this instruction. A score of 5 means the capability is very used, while 1 means it is not used at all.

- - - - - - - - - - - - - - - - - - - - - - - - - - - - - - - - - - - - - - - - - - - - - - - - - - - - -

**User Prompt**
## Task
You are given a user instruction and its reference response, along with a list of 20 capabilities. For each capability, rate the degree to which it is required to respond to this instruction.

## User Instruction
{input}

## Reference Response
{output}

## Capabilities
{capability_list}

## Requirements
- For each capability, provide an integer **score from 1 to 5**. A score of 5 means the capability is very used, while 1 means it is not used at all.
- Include a brief **reasoning** for each score, explaining how you determined the score.
- Output the result in **JSON format** as follows:

```json
{
  "1": {"reasoning": "THE REASONING", "score": SCORE},
  "2": {"reasoning": "THE REASONING", "score": SCORE},
  "3": {"reasoning": "THE REASONING", "score": SCORE},
  ...
}
```

- Do NOT include any additional text outside of the JSON format, as **I will directly use 'json.loads' in Python to convert your output to a dictionary object**.

Table 20: The scoring prompt for WildChat10K (Zhao et al., 2024a) used by QUALEVAL.

---

**System Prompt**
Given a Python coding problem with its correct implementation and a numbered list of capabilities, rate each capability on a scale of 1-5 to indicate its relevance in solving this problem. A score of 5 means the capability is very used, while 1 means it is not used at all.

- - - - - - - - - - - - - - - - - - - - - - - - - - - - - - - - - - - - - - - - -

**User Prompt**
## Task
You are given a Python coding problem and its correct implementation, along with a list of 20 capabilities. For each capability, rate the degree to which it is required to solve this problem.

## Coding Problem
{input}

## Correct Implementation
{output}

## Capabilities
{capability_list}

## Requirements
- For each capability, provide an integer **score from 1 to 5**. A score of 5 means the capability is very used, while 1 means it is not used at all.
- Include a brief **reasoning** for each score, explaining how you determined the score.
- Output the result in **JSON format** as follows:

```json
{
  "1": {"reasoning": "THE REASONING", "score": SCORE},
  "2": {"reasoning": "THE REASONING", "score": SCORE},
  "3": {"reasoning": "THE REASONING", "score": SCORE},
  ...
}
```

- Do NOT include any additional text outside of the JSON format, as **I will directly use 'json.loads' in Python to convert your output to a dictionary object**.

Table 21: The scoring prompt for DS-1000 (Lai et al., 2023) used by QUALEVAL.

**System Prompt**
Given a mathematical question and its correct solution, check whether the provided mathematics skill or capability is required by the key aspects of the solution.

- - - - - - - - - - - - - - - - - - - - - - - - - - - - - - - - - - - - - - - - - - - - - -

**User Prompt**
## Question
{input}

## Solution
{output}

## Skill or Capability
{capability}

## Requirement
If the provided mathematics skill or capability is required by the key aspects of the solution, output YES. Otherwise, output NO.
You should output either YES or NO with no additional text, otherwise, the output will NOT be accepted.

Table 22: The prompt for determining whether or not a given MATH (Hendrycks et al., 2021b) benchmark instance tests for a given capability.

**System Prompt**
Given a user instruction and a reference response to the instruction, check whether the provided skill or capability is required by the key aspects of responding to the instruction.

- - - - - - - - - - - - - - - - - - - - - - - - - - - - - - - - - - - - - - - - - - - - - -

**User Prompt**
## Instruction
{input}

## Response
{output}

## Skill or Capability
{capability}

## Requirement
If the provided skill or capability is required by the key aspects of responding to the instruction, output YES. Otherwise, output NO.
You should output either YES or NO with no additional text, otherwise, the output will NOT be accepted.

Table 23: The prompt for determining whether or not a given WildChat10K (Zhao et al., 2024a) benchmark instance tests for a given capability.

**System Prompt**
Given a Python coding problem (involving data science) and its correct Python implementation, check whether the provided coding skill or capability is required by the key aspects of the implementation.

- - - - - - - - - - - - - - - - - - - - - - - - - - - - - - - - - - - - - - - - - - - - - - - -

**User Prompt**
## Problem
{input}

## Implementation
{output}

## Skill or Capability
{capability}

## Requirement
If the provided coding skill or capability is required by the key aspects of the implementation, output YES. Otherwise, output NO.
You should output either YES or NO with no additional text, otherwise, the output will NOT be accepted.

Table 24: The prompt for determining whether or not a given DS-1000 (Lai et al., 2023) benchmark instance tests for a given capability.

for controlling the sampling process. Taking correctness-based accuracy as an example, for the $i$-th benchmark instance, we compute the probability of it being solved correctly (i.e., $\mathbb{P}[g_i = 1]$) as $p \times d^m$, where $m$ is the number of ground-truth weaknesses in $W^*$ for which the instance is an associated instance. Finally, we independently sample correctness (1 or 0) for each $g_i$ using these computed probabilities, resulting in a synthetic evaluation result (on the profiling set). By design, the ground-truth weakness profile $W^*$ exactly represents the real weaknesses for this generated synthetic evaluation result, as we were mimicking the evaluation behavior of a hypothetical LM with exactly these weaknesses. As we described above, when using correctness-based accuracy as the metric for MATH, $p \times d^m$ represents the probability of an instance's evaluation result being correct. Similarly, when using win-rate as the metric for WildChat10K, $p \times d^m$ denotes the probability of the (hypothetic) evaluated LM being preferred by the LM judge; specifically, we simulate the judge's preference by sampling twice, once for the original order of responses and once after swapping their order (see Appendix C). For each benchmark, we generated three synthetic evaluation results using the hyperparameters $p = 0.7$ and $d \in \{0.2, 0.4, 0.5\}$.

Given a weakness profile $W$ generated by a method, we measure its similarity to $W^*$. We define "Precision" as $\sum_{w_i \in W} |A(w_i) \cap (\cup_{w_j^* \in W^*} A(w_j^*))| / |A(w_i)| / |W|$ to measure desideratum 1, i.e., how precisely identified weaknesses align with ground-truth ones; similarly, we define "Recall" as $\sum_{w_j^* \in W^*} |A(w_j^*) \cap (\cup_{w_i \in W} A(w_i))| / |A(w_j^*)| / |W^*|$ to measure desideratum 2, i.e., how comprehensively ground-truth weaknesses are covered; finally, their harmonic mean, F1, provides a balanced measurement. By default, we use the profiling set itself as the test set for computing $A$ in the formulas above; we also show the results of using a separate test set distinct from the profiling set in Appendix E.3.3.

### E.3.2 Analysis on Experimental Results

This subsection provides additional analysis on the experimental results in Section 5.2.

To better understand why TEXTDIFF and QUALEVAL are outperformed, we show the Precision and Recall curves in Figure 11 and 12. These curves show that both methods suffer from poor Precision, indicating that the weaknesses they identify cannot precisely pinpoint where the LM fails. We present the identified weaknesses from TEXTDIFF, QUALEVAL, and

| Method | Weakness Profile |
|---|---|
| TEXTDIFF | Solving complex trigonometric equations and identities. (**11.11%**)
Handling and solving inequalities involving multiple variables. (**18.18%**)
Solving problems involving optimization and maximizing or minimizing expressions. (**15.79%**)
Understanding and applying properties of circles and their tangents. (**30.0%**)
Understanding and applying properties of vectors and vector operations. (**46.34%**)
Understanding and applying properties of matrices and determinants. (**60.0%**)
Handling and solving problems involving complex numbers and their operations. (**44.44%**)
Understanding and applying geometric transformations and properties. (**50.0%**)
Understanding and applying properties of polynomials and their roots. (**36.77%**)
Applying the Pythagorean theorem and properties of right triangles. (**41.46%**) |
| QUALEVAL | Applying optimization techniques and inequalities in problem-solving (**22.22%**)
Utilizing properties of geometric figures, including transformations and conic sections (**41.07%**)
Analyzing sequences, series, and their properties (**34.92%**)
Analyzing and solving inequalities and systems of equations (**37.31%**)
Calculating combinations, permutations, and applying counting principles (**47.54%**)
Applying vector operations and understanding geometric interpretations (**45.45%**)
Employing logical reasoning and problem-solving strategies (**48.89%**)
Calculating areas, volumes, and perimeters of geometric shapes (**28.57%**)
Understanding and manipulating complex numbers and their properties (**44.44%**)
Understanding and applying properties of functions, including logarithmic, exponential, and trigonometric functions (**34.57%**) |
| EVALTREE | Analyzing and applying geometric properties, relationships, and transformations across various contexts and configurations. (**37.71%**)
Analyzing and applying geometric reasoning to understand spatial relationships and calculate dimensions in two- and three-dimensional contexts. (**35.05%**)
Analyzing and applying recursive relationships and mathematical sequences to identify patterns and solve combinatorial problems. (**25.0%**)
Analyzing and manipulating numerical properties and representations across various numeral systems. (**46.53%**)
Analyzing and manipulating polynomial equations and their complex roots to evaluate relationships and distances. (**16.67%**)
Analyzing and optimizing geometric relationships using trigonometric principles and the Triangle Inequality. (**5.56%**)
Analyzing polynomial relationships and roots using Vieta's formulas and complex number properties. (**14.81%**)
Applying quadratic equations and trigonometric principles to solve for variable values and integer solutions. (**0.0%**)
Formulating, analyzing, and applying combinatorial reasoning to evaluate mathematical relationships and count objects under constraints. (**40.0%**)
Optimizing mathematical expressions and relationships through analysis, inequalities, and constraints. (**26.11%**) |

Table 25: Weakness profiles generated by TEXTDIFF, QUALEVAL, and EVALTREE, along with the LM performance on the associated instances (in the test set) of each identified weakness. Methods are run on Llama 3.1 8B Instruct (Dubey et al., 2024) evaluation result on MATH (Hendrycks et al., 2021b).

EVALTREE when the weakness profile size is 10 in Table 28, along with their corresponding Precision, Recall, and F1; they are based on applying the three methods to the synthetic evaluation result generated for the MATH benchmark, with the probability hyperparameters set to $p = 0.7$ and $d = 0.2$. We observe that EVALTREE achieves significantly higher Precision compared to the other two methods, while maintaining a quite high Recall, indicating that EVALTREE can more precisely pinpoint specific areas where the LM underperforms and thus better satisfy desideratum 1. For example, EVALTREE identified the weakness *"Analyzing and applying relationships among polynomial expressions and their roots using Vieta's formulas,"* which closely aligns with the ground-truth weakness *"Solving polynomial equations by analyzing relationships through Vieta's formulas;"* in contrast, TEXTDIFF and QUALEVAL identified two much coarser-grained weaknesses, *"Handling problems involving the properties of polynomials and their roots"* and *"Solving linear, polynomial, and quadratic equations, including factoring and roots"* respectively, failing to capture the critical aspect of Vieta's formulas.

This example shows the advantage of EVALTREE modeling the capabilities tested within a benchmark at diverse granularities. By contrast, QUALEVAL, relying on a single-level categorization, can only represent a fixed-granularity structure, which fails to sufficiently model the intricate and interrelated structure of capabilities tested within a benchmark. Consequently, it fails to capture the nuanced performance of LMs on fine-grained capabilities, leading to its inability to detect granular weaknesses. In contrast, EVALTREE successfully models the complexity of capabilities tested within a benchmark by the hierarchical structure of capability trees; this lets us analyze capabilities at varying granularities flexibly, from broad categories to specific skills. By incorporating this flexibility, EVALTREE captures much more detailed and comprehensive information about LM performance, so it can be superior.

| Index | Capability Description |
|-------|------------------------|
| 1 | Solving problems involving complex numbers and trigonometric identities, including the use of algebraic manipulation, polar forms, and exponentiation of complex numbers. |
| 2 | Analyzing combinatorial problems using counting principles and recurrence relations to count and analyze complex arrangements. |
| 3 | Applying geometric formulas to calculate areas, volumes, and other properties of three-dimensional shapes. |
| 4 | Analyzing numbers using prime factorization to solve problems involving divisibility and coprimality. |
| 5 | Solving probability problems using geometric probability. |
| 6 | Solving polynomial equations by analyzing relationships through Vieta's formulas. |
| 7 | Using trigonometric identities and polynomial identities to reduce complex expressions. |
| 8 | Involving geometric partitioning or area considerations to calculate probabilities. |
| 9 | Analyzing quadratic inequalities through factoring. |
| 10 | Applying the properties of divisibility to find common factors using the Greatest Common Divisor. |

Table 26: The manually curated ground-truth weakness profile for MATH (Hendrycks et al., 2021b), used in Ground-Truth Weakness Assessment (Section 5.2).

### E.3.3 Computing F1 on a Separate Set

In this subsection, we present the results of Section 5.2 using a separate test set (distinct from the profiling set) for computing $A$ in the formulas provided in Appendix E.3.1.

Here, for the MATH benchmark (Hendrycks et al., 2021b), the test set is its released training set (consisting of 7,500 instances). For WildChat10K, we sample another 10K instances from WildChat (Zhao et al., 2024a) as the test set, using the same construction process as the profiling set (WildChat10K) and ensuring no overlap with WildChat10K by excluding previously included instances. The results, shown in Figure 13, demonstrate consistent observations with those observed on the original results in Figure 4.

| Index | Capability Description |
|---|---|
| 1 | Proficiency in designing intuitive, user-friendly interfaces. |
| 2 | Proficiency in editing and proofreading for academic papers. |
| 3 | Financial forecasting and risk analysis. |
| 4 | Proficiency in understanding and/or utilizing object-oriented programming concepts. |
| 5 | Game mechanics design and balancing. |
| 6 | Crisis communication management by media response crafting. |
| 7 | Synthesis of statistical analysis and data interpretation for business purposes. |
| 8 | Helping the users with their own mental health. |
| 9 | Evaluating complex moral dilemmas and proposing socially responsible solutions. |
| 10 | Event planning by logistical coordination. |

Table 27: The manually curated ground-truth weakness profile for WildChat10K (Zhao et al., 2024a), used in Ground-Truth Weakness Assessment (Section 5.2).

### E.4 Experimental Details of Extrinsic Assessment

This section provides additional details about Section 5.3.

We use OpenAI's `gpt-4o-mini-2024-07-18` (OpenAI, 2024a) in our experiments to generate (synthetic) data inputs; the input generation prompts for MATH (Hendrycks et al., 2021b) and DS-1000 (Lai et al., 2023) are provided in Table 29 and Table 30, respectively; we set the max new tokens and temperature to 4096 and 1.0 (for generation diversity), respectively. We also use `gpt-4o-mini-2024-07-18` to generate outputs for each collected input; the output generation prompts for MATH and DS-1000 are provided in Table 31 and Table 32, respectively; we set the max new tokens and temperature to 4096 and 0.0, respectively.

For the generic-capability-guided data collection strategy, we use a description of the benchmark's overall targeted capability as guidance (in the input generation prompt) for synthetic data generation. The descriptions are *"General mathematical reasoning capability across Prealgebra, Algebra, Number Theory, Counting and Probability, Geometry, and Intermediate Algebra."* and *"General Python coding capability across data science libraries: NumPy, Pandas, TensorFlow, PyTorch, SciPy, Scikit-learn, and Matplotlib."* for MATH and DS-1000, respectively.

For the EVALTREE-guided data collection strategy, we set the accuracy threshold $\tau$ to 0.4 in the node extraction algorithm described in Section 3.2. This resulted in 9 identified weaknesses for MATH and 5 for DS-1000; the same number of weaknesses was identified when using the TEXTDIFF-guided strategy and the QUALEVAL-guided strategy, ensuring that all weakness-guided data collection strategies use weakness profiles of the same size. When sampling five in-context examples for input generation given an identified weakness in a weakness-guided data collection strategy, the examples are sampled from the associated instances (in the profiling set) of the identified weakness in the TEXTDIFF-guided strategy, from the instances assigned to the identified weakness in the QUALEVAL-guided strategy, and from the instances linked to the corresponding node in the EVALTREE-guided strategy.

We provide an example of synthetic data inputs generated for Llama 3.1 8B Instruct on MATH. One EVALTREE-identified weakness is *"Analyzing and optimizing geometric relationships using trigonometric principles and the Triangle Inequality."* A synthetic data input generated under the guidance of this weakness is *"In triangle ABC, the lengths of sides AB and AC are 15 cm and 20 cm, respectively. If angle A measures 60°, what is the length of side BC rounded to the nearest whole number?"* In contrast, a synthetic data input guided by the generic capability is *"A trader bought a certain number of apples for \$0.75 each and then sold them for \$1.00 each. If he had a total profit of \$15 after selling all the apples, how many apples did he sell?"* This example highlights that EVALTREE provides targeted guidance for data collection.

For each data collection strategy, we collect 128 instance inputs for training. We finetune the models using LoRA (Hu et al., 2022), with a rank of 256, an alpha of 512, and a dropout rate of 0.1. The batch size is fixed at 8, and the maximum sequence length is set to 1024

| Method | Weakness Profile |
|---|---|
| TEXTDIFF | Solving problems involving the properties of prime numbers and their factorizations
Solving equations involving trigonometric identities and simplifications
Handling complex numbers and their operations
Solving problems involving combinatorics and permutations
Applying the Law of Cosines and Law of Sines in non-right triangles
Handling problems involving the calculation of probabilities and combinatorial counting
Handling problems involving the calculation of areas and volumes of geometric shapes
Handling problems involving the properties of polynomials and their roots
Understanding and applying the properties of quadratic equations and their roots
Handling problems involving divisibility and modular arithmetic |
| QUALEVAL | Understanding and applying number theory concepts, including prime factorization and modular arithmetic
Understanding and manipulating complex numbers and their properties
Calculating combinations, permutations, and applying counting principles
Calculating areas, volumes, and perimeters of geometric shapes
Calculating probabilities and utilizing statistical methods for data analysis
Employing logical reasoning and problem-solving strategies
Understanding and applying properties of functions, including logarithmic, exponential, and trigonometric functions
Solving linear, polynomial, and quadratic equations, including factoring and roots
Applying optimization techniques and inequalities in problem-solving
Analyzing and solving inequalities and systems of equations |
| EVALTREE | Simplifying and solving trigonometric and complex expressions using algebraic manipulation, identities, and properties of periodic functions
Manipulating complex numbers and applying series and binomial techniques to derive geometric properties
Analyzing and calculating complex numbers through polar coordinates, polynomial equations, and algebraic manipulation
Analyzing and applying relationships among polynomial expressions and their roots using Vieta's formulas
Solving and manipulating algebraic, quadratic, and probability equations
Analyzing and applying prime factorization, divisibility, and the relationships between greatest common divisors and least common multiples to solve mathematical problems
Analyzing and calculating prime factorization and divisibility within factorials
Analyzing and calculating prime numbers and whole numbers through factorization and divisor techniques
Factoring integers and polynomials to analyze prime components, apply properties of exponents, and identify valid combinations
Calculating and analyzing geometric properties and volumes of three-dimensional shapes using formulas and algebraic manipulation |

Table 28: Weakness profiles generated by TEXTDIFF, QUALEVAL, and EVALTREE. TEXTDIFF achieves a Precision of 0.4787, a Recall of 0.9450, and an F1 of 0.6355. QUALEVAL achieves a Precision of 0.3494, a Recall of 0.9975, and an F1 of 0.5175. EVALTREE achieves a Precision of 0.7064, a Recall of 0.8081, and an F1 of 0.7538. Methods are run on the synthetic evaluation result generated for the MATH (Hendrycks et al., 2021b) benchmark, with $p = 0.7$ and $d = 0.2$. The ground-truth weakness profile is provided in Table 26.

---

**System Prompt**
You are a creative and logical assistant tasked with generating new mathematics questions. Your goal is to create a single, clear question aligned with a given mathematical capability.

- - - - - - - - - - - - - - - - - - - - - - - - - - - - - - - - - - - - - - - - - - - - - - - - -

**User Prompt**
## Task
Generate one unique mathematics question demonstrating the following capability:
{capability}

Please ensure the following:
- You will be given {instance_num} example questions for reference. Use the examples solely to understand the capability, NOT as templates, i.e., the generated question must not replicate, paraphrase, or directly resemble the example questions in structure, wording, or context.
- The question must ask for only one result, such as a numerical value, while adhering to logical constraints (e.g., quantities must be positive, and counts for people must be integers).

## Provided Examples
{example_inputs}

## Requirements
- Do NOT include a solution in the generated question.
- Ensure the question is plausible, reasonable, and relevant to the given capability.

---

Table 29: The (synthetic data) input generation prompt for MATH (Hendrycks et al., 2021b).

tokens. Training is conducted using BF16 precision. The optimizer is configured with a learning rate of 1E-4, a cosine learning rate scheduler, a warmup ratio of 0.1, and no weight decay. The models are trained for 3 and 2 epochs in the experiments on MATH and DS-1000, respectively. These configurations are applied consistently across all experiments.

### E.5 Details of LM Usage Costs

Let the number of benchmark instances (the size of profiling set) be denoted as $N$.

The main LM usage cost of EVALTREE is incurred during the Capability Annotation stage, where each instance requires one LM call, and the Capability Description stage, where each non-leaf node of the capability tree also requires one LM call. The cost of the sentence embedding model used in the Capability Embedding stage is negligible in comparison. As the number of non-leaf nodes in the capability tree is smaller than $N$, the total number of LM calls and thus the overall LM usage cost for EVALTREE scale as $O(N)$.

For TEXTDIFF, the main LM usage cost is incurred when determining the associated instances for each potential weakness outputted by the diagnostic LM. Each potential weakness requires $O(N)$ LM calls, causing the total number of LM calls and thus the overall LM usage cost to scale linearly with the number of potential weaknesses outputted by the diagnostic LM, which is the upper bound of the weakness profile size.

For QUALEVAL, the main LM usage cost comes from scoring each pair of benchmark instances and capabilities derived from all benchmark instances. The scoring LM generates a natural language reasoning for each score (see prompts in Appendix D.2), making the output token cost a significant component of the total cost. Since the length of the LM's output scales linearly with the predefined number of capabilities (which is the upper bound of the weakness profile size), the overall LM usage cost (roughly) scales accordingly.

**System Prompt**
You are a creative and logical assistant tasked with generating new Python programming problems. Your goal is to create a single, clear problem aligned with a given data science capability.

- - - - - - - - - - - - - - - - - - - - - - - - - - - - - - - - - - - - - - - - - - -

**User Prompt**
## Task
Generate one unique Python programming problem demonstrating the following capability:
{capability}

Please ensure the following:
- You will be given {instance_num} example problems for reference. Use the examples solely to understand the capability and the desired problem format. The generated problem must not replicate, paraphrase, or directly resemble the example problems in structure, wording, or context.
- The problem must ask for one piece of Python code that fills in a blank, ensuring clarity and conciseness while being grounded in real-world data science scenarios.

## Provided Examples
{example_inputs}

## Requirements
- Do NOT include a solution in the generated problem. Please output the generated problem directly, without any additional text, explanation, or commentary.
- Ensure the problem is plausible, reasonable, and relevant to the given capability.
- Adhere to logical programming constraints, such as correct syntax and realistic data or outcomes.

Table 30: The (synthetic data) input generation prompt for DS-1000 (Lai et al., 2023).

**System Prompt**
You are a precise and logical assistant. Solve the following mathematics problem step by step, explaining each step clearly.
Enclose the final answer to the mathematics question within \boxed{}.

- - - - - - - - - - - - - - - - - - - - - - - - - - - - - - - - - - - - - - - - - - -

**User Prompt**
{input}

Table 31: The output generation prompt for MATH (Hendrycks et al., 2021b).

**System Prompt**
Write a short code following the given format and indentation. Place the executable code between  and  tags, without any other non-executable things. Please provide ONLY the code completion needed. Do NOT repeat the context code.

- - - - - - - - - - - - - - - - - - - - - - - - - - - - - - - - - - - - - - - - - - -

**User Prompt**
{input}

Table 32: The output generation prompt for DS-1000 (Lai et al., 2023).

As analyzed above, the scale coefficients of TEXTDIFF and QUALEVAL grow linearly with the (maximum) weakness profile size, making their costs significantly higher than EVALTREE, which maintains a linear cost scaling with the number of benchmark instances regardless of the weakness profile size. This difference makes EVALTREE substantially more cost-efficient in terms of LM usage cost, especially when the weakness profile size is large.

## F   Quantitative Analysis of Flaws in Chatbot Arena's Evaluation Practice

This section provides additional quantitative analysis of the flaws in Chatbot Arena's human-voter-based evaluation practice, discussed in Section 6. We use the OpenAI Moderation API[10] with the model omni-moderation-2024-09-26 to assess toxicity in the following; this is a tool that evaluates whether or not a given text contains toxic content.

We first examine the user instructions for instances linked to the node *"Facilitating inclusive, ethical, and strategic communication and engagement across diverse and sensitive contexts"*. Across the entire Chatbot Arena benchmark, 4.72% of instances have toxic user instructions; however, at this specific node, the proportion rises sharply to 19.50%. It is worth noting that people found that the OpenAI Moderation API may have a low recall (Zhao et al., 2024a), resulting in numerous false negatives (toxic instructions not flagged as such), so the actual proportion of toxic user instructions should be higher. Despite this limitation, the observed toxicity rate at this node is significantly higher than the benchmark average, confirming that it contains a disproportionate number of user instructions with toxic requests, which aligns with the natural language description of the capability represented by the node.

We then examine the trend of human voter preferences when comparing two responses, one providing a toxic response and the other providing a non-toxic response (often by refusing to answer). We focus on human comparison pairs where one response is flagged as toxic and the other is not. Across all such comparison pairs, the proportion where the toxic response is preferred is 50.89%; when also counting "tie" cases to consider all cases where the non-toxic response is not preferred, the proportion rises to 71.98%. This issue is even more serious at the node *"Facilitating inclusive, ethical, and strategic communication and engagement across diverse and sensitive contexts"*; among comparison pairs for the node's instructions, these two numbers rise significantly to 86.84% and 97.37%, respectively. These results confirm the observation that human voters tend to prefer toxic responses (that do not refuse to answer), diverging from the intended values. They underscore the need for careful refinement of evaluation practices to ensure alignment with the desired principles.

## G   Ablation Study: Alternative Approach to Tree Construction

In this section, we explore an alternative approach to the tree construction pipeline introduced in Section 3.1. In this approach, we still follow the four-stage pipeline. For the **stage (3)**, instead of recursively building the hierarchical structure in a top-down, recursive way, we use the hierarchical clustering algorithm (Müllner, 2011), implemented with scipy.cluster.hierarchy.linkage[11]. The other stages remain unchanged. We did not adopt this approach because it always produces a binary tree, where the optimal number of each node's children could be more than two and diverse; a binary tree cannot meet this need, whereas our default approach can automatically determine a (potentially) optimal number of children at each node. We also empirically observed that trees constructed by hierarchical clustering sometimes have unbalanced structures; for example, the left subtree of the root may contain very few instances while the right subtree contains many.

We compare EVALTREE using the default capability tree construction pipeline with EVAL-TREE using the capability tree built with the hierarchical clustering algorithm in the experi-

---

[10]https://platform.openai.com/docs/api-reference/moderations
[11]https://docs.scipy.org/doc/scipy/reference/generated/scipy.cluster.hierarchy.linkage.html. The method is set to average, the metric to cosine, and all other hyperparameters are set to their default values.

|  | MATH | DS-1000 |
|---|---|---|
| Initial LM | 48.70 | 29.20 |
| EVALTREE | **52.42(±0.28)** | **36.90(±0.34)** |
| EVALTREE (Hierarchical Clustering) | **52.88(±0.65)** | 33.36(±0.36) |

Table 33: Accuracy (%) of different LMs on MATH and DS-1000 test sets. The initial LM is Llama 3.1 8B Instruct (Dubey et al., 2024) for MATH and DeepSeek-Coder-Base 6.7B (Guo et al., 2024) for DS-1000, respectively. See Section 5.3 for the experimental setup. We compare EVALTREE using the default capability tree construction pipeline with EVALTREE using the capability tree built with the hierarchical clustering algorithm here. Synthetic data (used to train the initial LM) are generated under the guidance of the weakness profiles produced by the two versions of EVALTREE, respectively. The accuracy (of a trained LM) is reported as mean±stderr ("stderr" refers to standard error) across five random seeds.

mental setup of Sections 5.1, 5.2 and 5.3. The results, shown in Figure 14 and 15 and Table 33, show that the default version outperforms the hierarchical-clustering-based version.

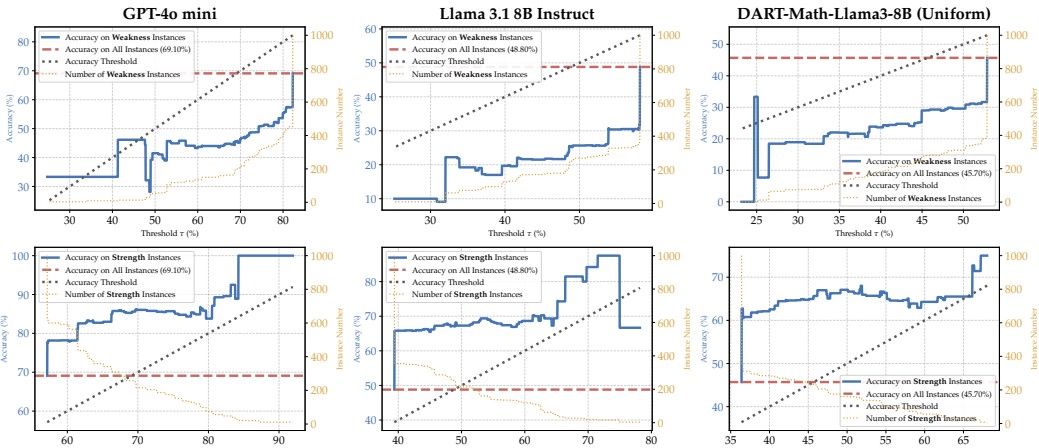

Figure 6: Accuracy curves of weakness instances and strength instances (from the test set) extracted using the random profiling/test split of the MATH benchmark (Hendrycks et al., 2021b). Experiments were conducted with GPT-4o mini (OpenAI, 2024a), Llama 3.1 8B Instruct (Dubey et al., 2024), and DART-Math-Llama3-8B (Uniform) (Tong et al., 2024). "All Instances" in the legend refers to all instances in the test set. A $y = x$ line is included in all figures to indicate the threshold $\tau$. The number of weakness/strength instances is shown as a reference; when the number is very low, the curve may exhibit significant fluctuations, affecting the general trend.

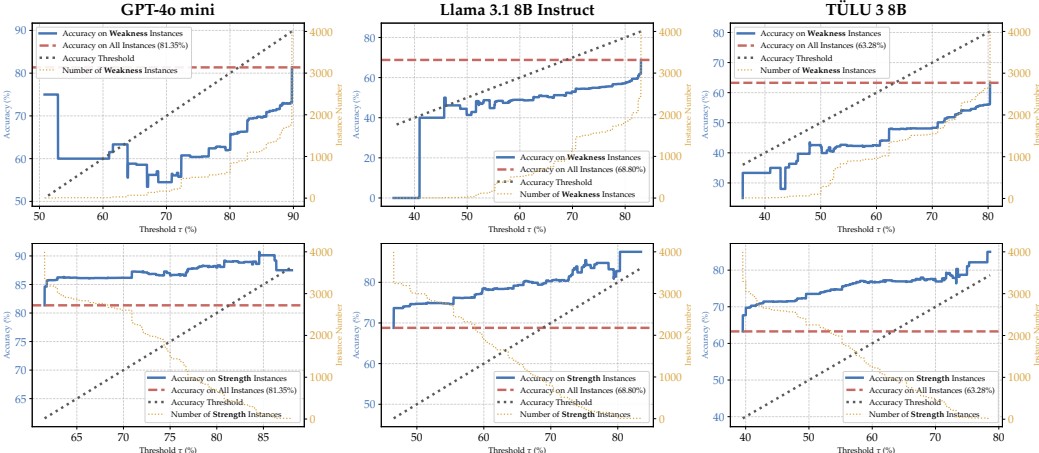

Figure 7: Accuracy curves of weakness instances and strength instances (from the test set) extracted using the random profiling/test split of the MMLU benchmark (Hendrycks et al., 2021a). Experiments were conducted with GPT-4o mini (OpenAI, 2024a), Llama 3.1 8B Instruct (Dubey et al., 2024), and TÜLU 3 8B (Lambert et al., 2024). "All Instances" in the legend refers to all instances in the test set. A $y = x$ line is included in all figures to indicate the threshold $\tau$. The number of weakness/strength instances is shown as a reference; when the number is very low, the curve may exhibit significant fluctuations, affecting the general trend.

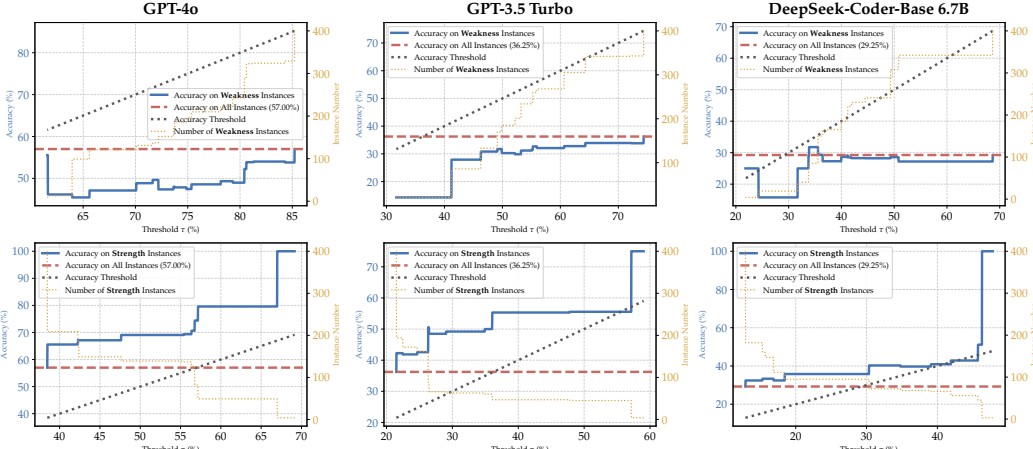

Figure 8: Accuracy curves of weakness instances and strength instances (from the test set) extracted using the random profiling/test split of the DS-1000 benchmark (Lai et al., 2023). Experiments were conducted with GPT-4o (OpenAI, 2024b), GPT-3.5 Turbo (OpenAI, 2022), and DeepSeek-Coder-Base 6.7B (Guo et al., 2024). "All Instances" in the legend refers to all instances in the test set. A $y = x$ line is included in all figures to indicate the threshold $\tau$. The number of weakness/strength instances is shown as a reference; when the number is very low, the curve may exhibit significant fluctuations, affecting the general trend.

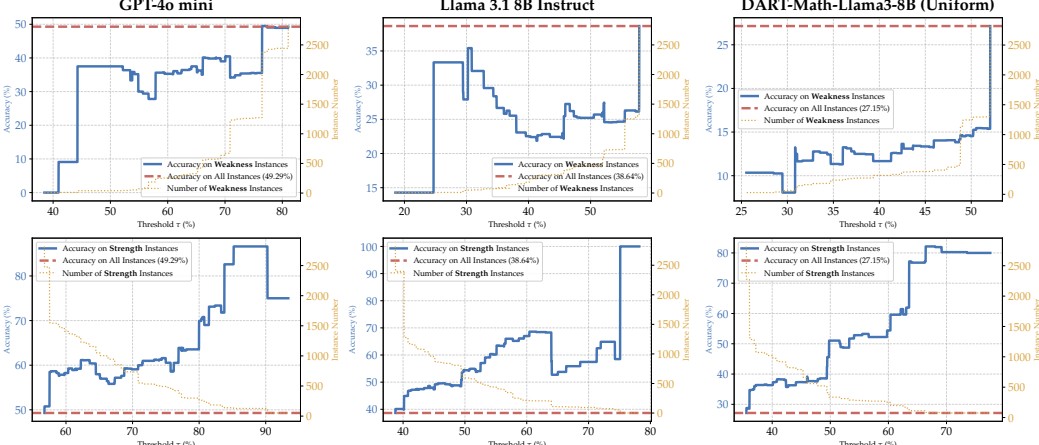

Figure 9: Accuracy curves of weakness instances and strength instances (from the test set) extracted using the MATH benchmark (Hendrycks et al., 2021b) as the profiling set and the CollegeMath benchmark (Tang et al., 2024) as the test set. Experiments were conducted with GPT-4o mini (OpenAI, 2024a), Llama 3.1 8B Instruct (Dubey et al., 2024), and DART-Math-Llama3-8B (Uniform) (Tong et al., 2024). "All Instances" in the legend refers to all instances in the test set. Note that the $y = x$ line of the threshold $\tau$ used in the node extraction algorithm is not drawn here, as comparing accuracies with the threshold directly is not meaningful due to the differing distributions of the profiling and test sets, which are from two different benchmarks. The number of weakness/strength instances is shown as a reference; when the number is very low, the curve may exhibit significant fluctuations, affecting the general trend.

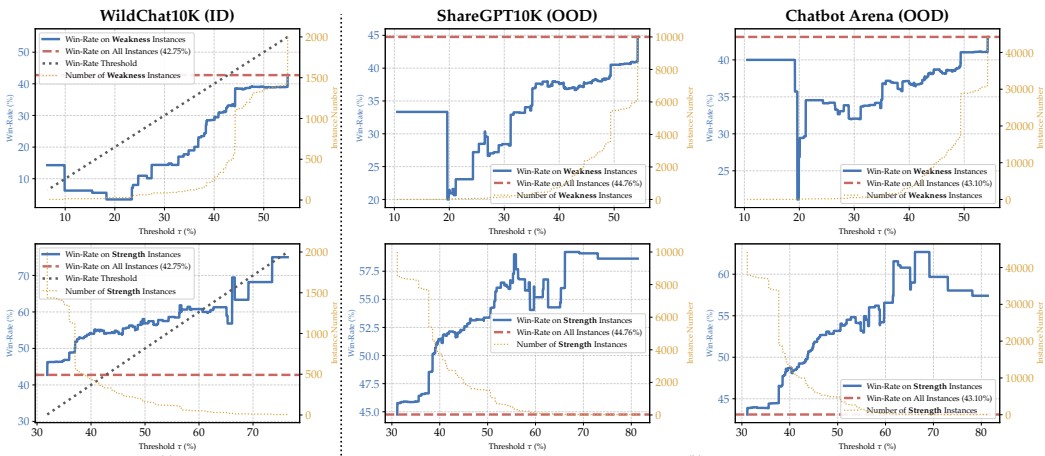

Figure 10: (a) Win-rate curves of weakness instances and strength instances (from the test set) extracted using the random profiling/test split of the WildChat10K benchmark (Zhao et al., 2024a). (b) Win-rate curves of weakness instances and strength instances (from the test set) extracted using the WildChat10K benchmark as the profiling set, with the ShareGPT10K and Chatbot Arena (Chiang et al., 2024) benchmarks serving as the respective test sets. The win-rate refers to the win-rate of Llama 3.2 3B Instruct (Meta, 2024) compared to Gemma 2 IT 2B (Rivière et al., 2024), as evaluated by the LM judge (Zheng et al., 2023; Dubois et al., 2023). "ID" indicates that the profiling and test sets are from the same benchmark (WildChat10K), whereas "OOD" indicates that they are from different benchmarks. The number of weakness/strength instances is shown as a reference; when the number is very low, the curve may exhibit significant fluctuations, affecting the general trend.

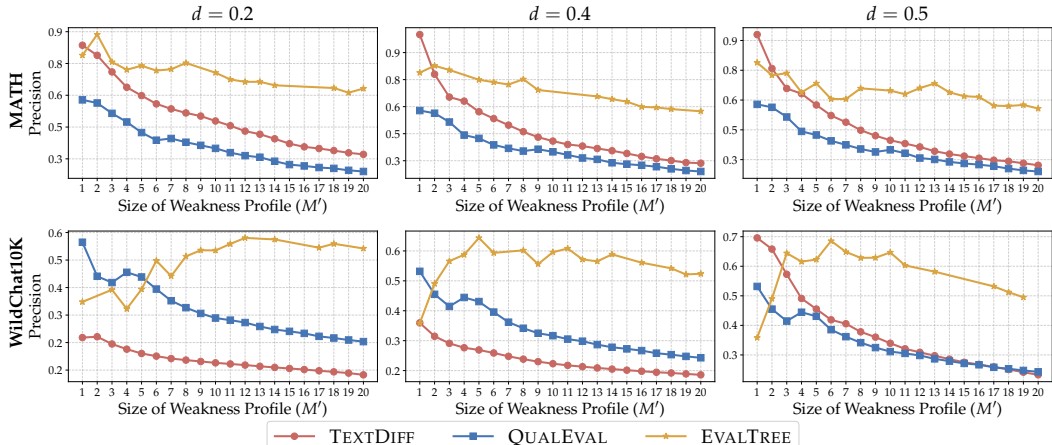

Figure 11: Precision score curves of TEXTDIFF, QUALEVAL, and EVALTREE, with the weakness profile size varying from 1 to 20. $d$ is a hyperparameter to control the sampling probability (see Appendix E.3.1).

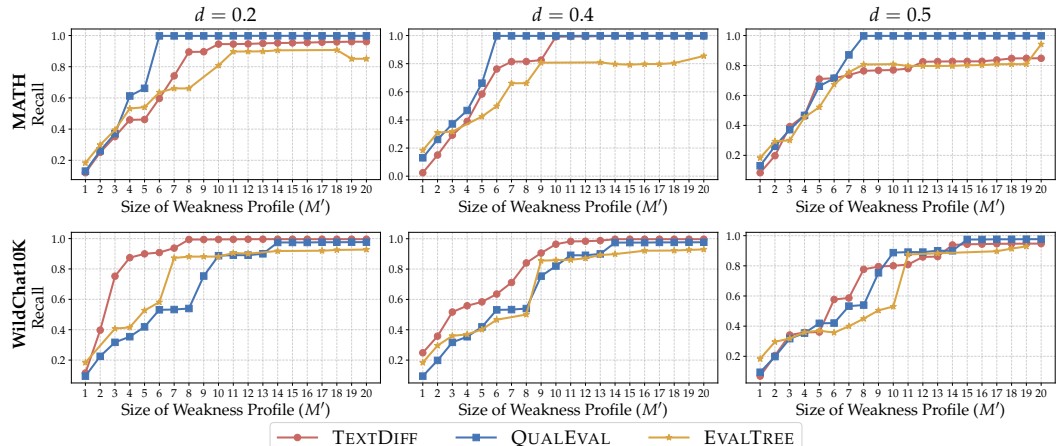

Figure 12: Recall score curves of TEXTDIFF, QUALEVAL, and EVALTREE, with the weakness profile size varying from 1 to 20. $d$ is a hyperparameter to control the sampling probability (see Appendix E.3.1).

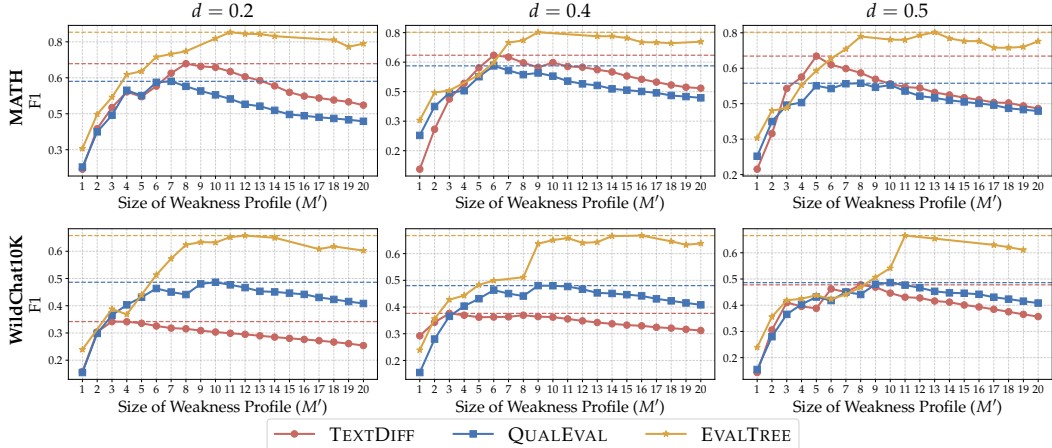

Figure 13: F1 score curves of TEXTDIFF, QUALEVAL, and EVALTREE, with the weakness profile size varying from 1 to 20. Precision, Recall, and thus F1 (more specifically, $A$ in the formulas provided in Appendix E.3.1) are computed on a separate test set, distinct from the profiling set used to generate the synthetic evaluation results. A horizontal line indicates each method's highest score. $d$ is a hyperparameter to control the sampling probability.

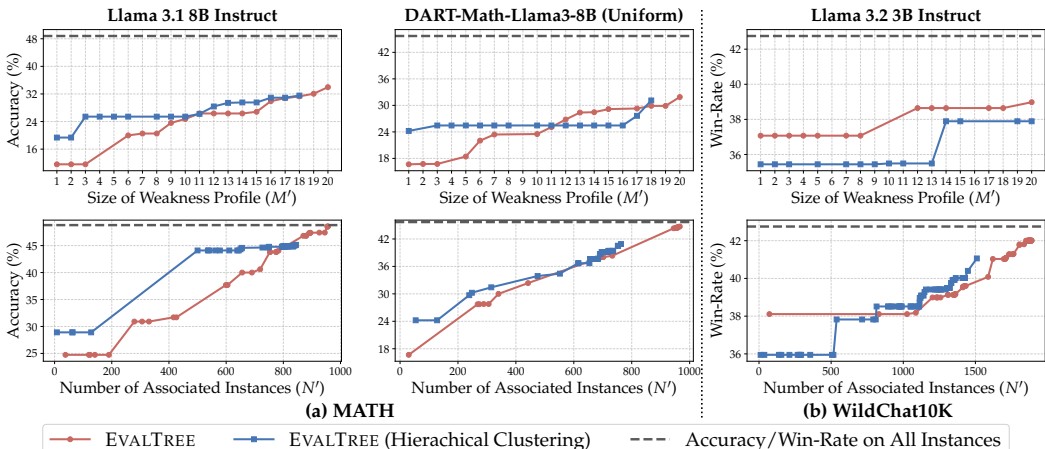

Figure 14: Curves of $\min\{\sum_{w_i \in W_\tau} F(A(w_i))/|W_\tau| \mid \forall\tau, |W_\tau| \geq M'\}$ (the first row) and $\min\{F(S_\tau) \mid \forall\tau, |S_\tau| \geq N'\}$ (the second row). See Section 5.1 for the experimental setup. Experiments in (a) were conducted on MATH with Llama 3.1 8B Instruct (Dubey et al., 2024) and DART-Math-Llama3-8B (Uniform) (Tong et al., 2024), and experiments in (b) were conducted on WildChat10K, where the win-rate is the percentage of instances in which Llama 3.2 3B Instruct (Meta, 2024) is preferred over Gemma 2 IT 2B (Rivière et al., 2024). We compare EVALTREE using the default capability tree construction pipeline with EVALTREE using the capability tree built with the hierarchical clustering algorithm here.

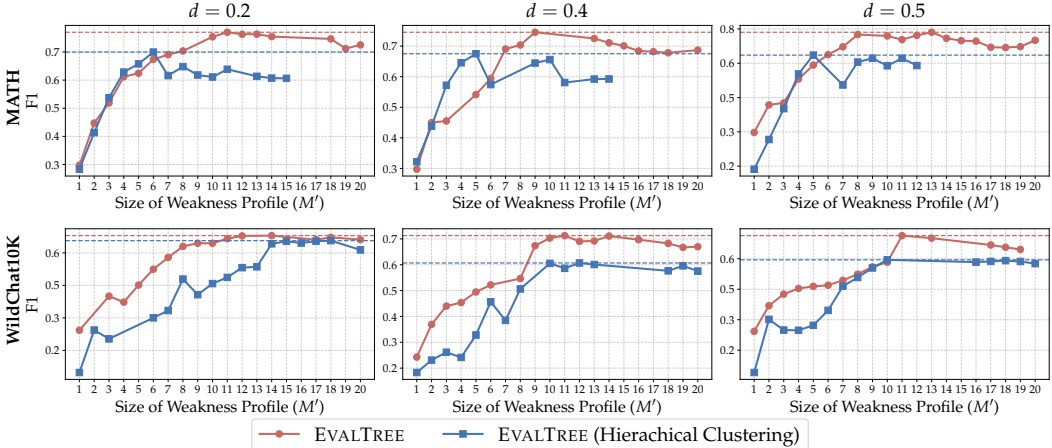

Figure 15: F1 score curves of EVALTREE using two different capability tree construction pipelines, with the weakness profile size varying from 1 to 20. See Section 5.2 for the experimental setup. A horizontal line indicates each method's highest score. $d$ is a hyperparameter to control the sampling probability (see Appendix E.3.1). We compare EVALTREE using the default capability tree construction pipeline with EVALTREE using the capability tree built with the hierarchical clustering algorithm here.

