# OpenReview forum: "EvalTree: Profiling Language Model Weaknesses via Hierarchical Capability Trees"
_colmweb.org/COLM/2025/Conference — COLM 2025_

### Official Review · Reviewer_pMeM · 2025-04-18

**Rating:** 10
**Confidence:** 4
**Ethics Flag:** 1

**Summary:**

This paper presents a highly intriguing research direction, namely the generation of LLM weakness profiles, which not only facilitates fine-grained analysis of LLM weaknesses but also enables targeted enhancement. To address this task, the authors propose an EvalTree framework for the automated synthesis of LLM weakness profiles from any chosen benchmarks. The study includes extensive experimental work and analysis to validate the effectiveness of EvalTree, and the final analysis on the chat arena proves particularly insightful and engaging.

**Questions To Authors:**

None

**Reasons To Accept:**

1. The research presented in this paper is highly compelling and holds significant value for both academia and industry.​​
​2. The methodology employed in this study is straightforward yet effective.​​
​3. Extensive experimental analyses validate the efficacy of the proposed approach, yielding numerous insightful findings and perspectives.​​

**Reasons To Reject:**

From my personal perspective, this paper exhibits no apparent flaws. I particularly appreciate both the research problem it addresses and the methodological approach it proposes.​​

---

> ### Author Response · Authors · 2025-05-31
>
> We are grateful for the highest recognition of the motivation behind EvalTree, the clarity and effectiveness of our methodology, and the extensiveness and depth of our experimental analysis (including some auxiliary experiments like the Chatbot Arena study). We thank Reviewer pMeM for the strongest support and extremely positive assessment of our work!

---

> > ### Comment · Reviewer_pMeM · 2025-06-04
> > **Response to the Authors**
> >
> > Thank you for your kind reply. I appreciate the authors’ engagement and wish them the best with their research.

---

### Official Review · Reviewer_n857 · 2025-05-07

**Rating:** 8
**Confidence:** 4
**Ethics Flag:** 1

**Summary:**

This paper introduces EvalTree, a method for profiling the weakness of a language model by automatically constructing a hierarchical capability tree over any benchmark and each node represents a capability described in natural language and is linked to a subset of instances that evaluate the specific capability.

The research demonstrates EvalTree's superiority over baseline methods in precision and comprehensiveness through experiments on benchmarks like MATH and WildChat. Furthermore, it shows the practical significance of this method by using EvalTree-identified weaknesses to guide more effective data collection for LM improvement  and by exposing flaws in existing evaluation practices like Chatbot Arena. The provision of an interactive interface further underscores its utility and potential for broader impact in understanding and enhancing LM capabilities.

A related but **not cited** work could be Domino [1], which also aims to identify and describe model failure modes.

Domino use cross-modal embeddings to discover and describe systematic error "slices" in broader AI applications (often vision/audio) via **natural language**, EvalTree is distinct in its hierarchical, capability-driven approach tailored for LMs. While both aim for interpretable error analysis, EvalTree focuses on structured, multi-level capability failures within LM tasks to guide improvement, whereas Domino leverages cross-modal concept understanding to identify coherent error-prone data segments.

[1] Domino: Discovering Systematic Errors with Cross-Modal Embeddings

**Reasons To Accept:**

1. This work tackles a critical challenge in AI: understanding why and where LMs fail, rather than just measuring overall performance. EvalTree provides a more precise, comprehensive, and actionable approach to diagnosing LM failures, offering human-interpretable insights that can directly aid developers in targeted model improvement.
2. EvalTree is novel in its hierarchical, capability-driven approach. The EvalTree methodology, with its four-stage tree construction pipeline (Capability Annotation, Capability Embedding, Recursive Clustering-Based Construction, and Capability Description), is clearly defined and technically sound.
3. The authors conduct comprehensive experiments on multiple benchmarks (e.g., MATH, WildChat) and LMs, employing their proposed quantitative assessments to compare EvalTree against relevant baselines (TextDiff, QualEval). The demonstrated success of weakness-guided data collection suggests a more efficient path to enhancing LM capabilities.
4. The application of EvalTree profiles to guide targeted data collection for LM improvement and to expose flaws in established evaluation practices like Chatbot Arena also represents original contributions. The public availability of an interactive interface for exploring capability trees further lowers the barrier for adoption and future research, promising a significant impact on how LMs are evaluated and iteratively improved.

**Reasons To Reject:**

1. Although the overall superiority of EvalTree over TextDiff and QualEval is shown, it is unclear to me how each design choice/element in EvalTree contributes to the final improvement. Does the improvement primarily stem from the hierarchical versus flat structure, or from other design choices? I would like to see more detailed ablation studies to gain a better understanding of the proposed method.
2. The ground-truth weakness assessment relies on synthetic LM evaluation results and manually curated weaknesses, which may not accurately reflect the complexity and subtlety of real-world LM failure modes. Is it possible to construct such ground truth on real-world datasets?
3. Additionally, the choice of baselines (TextDiff, QualEval), while representative, might omit comparisons with other potentially stronger or more relevant error analysis or slice discovery techniques that could provide a more challenging benchmark for EvalTree's performance. More baselines could further strengthen the paper [1, 2, 3].

[1] Domino: Discovering Systematic Errors with Cross-Modal Embeddings
[2] Discover, Explanation, Improvement: An Automatic Slice Detection Framework for Natural Language Processing
[3] Beyond Accuracy: Behavioral Testing of NLP models with CheckList

---

> ### Author Response · Authors · 2025-05-31
> **Rebuttal from Authors [1/n]**
>
> We sincerely thank Reviewer n857 for their thoughtful and constructive review. We are grateful for their recognition of EvalTree’s novelty, its comprehensive and interpretable approach to diagnosing LM weaknesses, the comprehensiveness of our experimental validation, and the practical value demonstrated through weakness-guided data collection and interactive interface. We also appreciate the helpful pointers to related work, and completely agree that better contextualization within the broader literature will further strengthen our contribution.
>
> Below, we address Reviewer n857’s concerns and suggestions in detail.
>
>
> + Reviewer n857 raises the question of how each design choice within EvalTree contributes to its overall effectiveness (e.g., if the hierarchical structure matters).
>   + We appreciate this question and agree that understanding the impact of different design choices is important. Many of EvalTree’s design choices follow naturally from its overarching goal; as such, most are relatively direct or standard (e.g., embedding instances before clustering). We have therefore focused our ablation efforts on the design choices that may require empirical validation.
>   + Regarding the importance of hierarchy versus flat categorization: We view this as a central axis of EvalTree’s design and thus a key factor of EvalTree’s success. A natural way to ablate this dimension is by comparing against existing methods that use a flat (single-level) categorization structure. Among such works, QualEval [1] and GoalEx [2] are the most relevant; in our experiments, we chose QualEval as a baseline for this comparison, as GoalEx involves extremely extensive LLM querying that becomes computationally infeasible at the benchmark scales we target. This comparison can be interpreted as an ablation of EvalTree’s hierarchical structure. Furthermore, in Appendix E.3.2, we provide qualitative analysis of this distinction: flat categorization can only operate at a fixed granularity, and thus fails to sufficiently model the intricate and interrelated structure of capabilities tested within a benchmark; in contrast, a hierarchical structure successfully models the complexity of capabilities tested within a benchmark, which lets us analyze capabilities at varying granularities flexibly, from broad categories to specific skills.
>   + As for how the hierarchical structure is constructed, EvalTree uses a recursive K-Means-based clustering algorithm by default. We ablate this choice in Appendix G by replacing it with a standard hierarchical clustering method. The results show that our default choice is more effective.
>   + We also conducted new ablation experiments to study how variations in capability annotations (from Stage 1) and capability embeddings (from Stage 2) might affect the resulting capability trees. Please see our overall response for further details.
>   + We appreciate the reviewer’s interest in understanding the contribution of individual components. We believe our analyses provide meaningful insights into the core design choices of EvalTree, and we thank the reviewer for the opportunity to clarify—we will make them more prominent in the paper.
> + Reviewer n857 expresses concern that the ground-truth weakness assessment relies on manually curated weaknesses rather than weaknesses derived from real-world model behavior.
>   + The reviewer is absolutely right to highlight the value of validating against real-world weaknesses. However, we argue that knowing such “ground-truth” weaknesses from real models is **currently NOT feasible**; and, in fact, this limitation is what motivates the need for formulating the research problem of weakness profiling and proposing methods like EvalTree in the first place. To reliably derive a ground-truth set of weaknesses from real models, one must already possess an accurate and comprehensive understanding of where and why the model fails, which is exactly the open research challenge that we hope to address.
>   + That said, we agree it is a limitation that the weaknesses used in our Ground-Truth Assessment are not “real” ones. We address this by relying on our full suite of weakness profiling method assessments.
>     + First, our Low-Performance Identification Assessment directly assesses how **precisely** each method can identify weaknesses using real model evaluation results.
>     + Second, to compare **comprehensiveness** with real model evaluation results, we rely on Extrinsic Assessment to provide an **indirect but reliable signal**, which examines how effectively the identified weaknesses guide training data collection. The intuition is as follows: if one method identifies a broader set of true model weaknesses, then collecting data to target those weaknesses should yield greater model performance gains.
>     + Ground-Truth Assessment provides a **direct signal** of how **precisely and comprehensively** each method can identify weaknesses (using synthetic model evaluation results).

---

> > ### Author Response · Authors · 2025-05-31
> > **Rebuttal from Authors [1/n, n=2]**
> >
> > + Reviewer n857 points out additional related work (Domino [3], Edisa [4], and CheckList [5]) that address the broader research goal of diagnosing model failures.
> >   + We appreciate these pointers and agree that they contribute meaningfully to the landscape of interpretable model analysis. While COLM does not permit changes to the paper during the rebuttal phase, we will be sure to incorporate discussion of these works into our Related Work section in the next revision.
> >   + That said, we also believe **these three methods are not applicable as methods for our weakness profiling problem**.
> >     + Domino and DEIM assume that the tasks have a closed output space, such as classification, where the model selects from a fixed set of labels, so this makes them more suited to traditional NLP tasks; in contrast, our weakness profiling problem does not hold such an assumption, and our assessment setup targets tasks with open-ended, free-form outputs, which we believe better reflects how LLMs are used by real-world users. Furthermore, both Domino and DEIM assume access to a finite, often predefined set of natural language descriptions for categories, e.g., photo topics in Domino or linguistic features in DEIM, while our problem setup does not provide such a predefined set. Due to these two reasons, Domino and DEIM cannot be run in our problem setup.
> >     + CheckList differs even more fundamentally from our setting. It uses a matrix of general linguistic capabilities and test types to guide evaluation, requiring practitioners to design or generate test cases targeting a predefined set of capabilities, treating these capabilities as the output space of weaknesses. In contrast, our weakness profiling problem aims to discover weaknesses without assuming any predefined set. As such, while CheckList is a valuable framework for manual, hypothesis-driven behavioral testing, it is not applicable as a method for our problem setup.
> >
> > ### Reference
> >
> > [1] [QualEval: Qualitative Evaluation for Model Improvement](https://arxiv.org/abs/2311.02807)
> >
> > [2] [Goal-Driven Explainable Clustering via Language Descriptions](https://arxiv.org/abs/2305.13749)
> >
> > [3] [Domino: Discovering Systematic Errors with Cross-Modal Embeddings](https://arxiv.org/abs/2203.14960)
> >
> > [4] [Discover, Explanation, Improvement: An Automatic Slice Detection Framework for Natural Language Processing](https://arxiv.org/abs/2211.04476)
> >
> > [5] [Beyond Accuracy: Behavioral Testing of NLP models with CheckList](https://arxiv.org/abs/2005.04118)

---

> ### Comment · Reviewer_n857 · 2025-06-06
>
> Thank you for the detailed and thoughtful rebuttal. I'm glad to see that the authors have clearly addressed my concerns, particularly regarding the design choices in EvalTree and the applicability of related work. I appreciate the clarifications and the new ablation results. Given these responses, I am satisfied with the changes and would like to raise my score.

---

> > ### Author Response · Authors · 2025-06-06
> >
> > Thanks again for your time and thoughtful, encouraging feedback. Your constructive suggestions are very helpful!

---

### Official Review · Reviewer_DECS · 2025-05-12

**Rating:** 8
**Confidence:** 3
**Ethics Flag:** 1

**Summary:**

The paper proposes an evaluation method to generate a weakness profile tree for LLMs. This helps in understanding what type of new data to collect to improve the model. The enhanced data collection process is shown to improve model performance is comparison to baselines.

The main motivation is that an evaluation benchmark is heterogenous and is meant to assess diverse model capabilities and varying granularities. Using a single aggregate metric doesn't reflect a diverse and detailed evaluation. This is a limitation of current model evaluations.

**Reasons To Accept:**

The paper proposes a well thought out an implemented solution to investigate model performance on the diverse data points in a benchmark. The proposed solution EvalTree is compared to existing baselines like TextDiff and QualEval and the reasons for developing this solution as an improvement over related work is well motivated. All the experimental details for the tree creation are provided which is a big plus for reproducibility. I hope the code for tree creation is made public. The compute cost comparison with existing baselines is favorable for the proposed solution.

**Reasons To Reject:**

No strong reasons for rejection.

---

> ### Author Response · Authors · 2025-05-31
>
> We appreciate the very high recognition of EvalTree’s motivation, its improvements over prior work, and the clarity of our experimental setup. We sincerely thank Reviewer DECS for the thoughtful and very positive review!
>
> For the release of code and data, please see our overall response for further details.

---

> > ### Comment · Reviewer_DECS · 2025-06-08
> >
> > Thanks for sharing the code anonymously and making the new experiment updates to the interactive demo. Looks good.

---

### Official Review · Reviewer_SQGa · 2025-05-13

**Rating:** 9
**Confidence:** 4
**Ethics Flag:** 1

**Summary:**

This paper introduces EVALTREE, a method for weakness profiling of LMs by constructing hierarchical capability trees. The central aim is to move beyond aggregate benchmark scores and provide interpretable, actionable insights into where a model fails and how it can be improved. Each node in the capability tree represents a natural-language-described capability, linked to a subset of benchmark instances. EVALTREE identifies nodes with statistically significant underperformance to produce a weakness profile.
The method is novel and ambitious, combining prompting-based capability labeling, sentence embeddings, recursive clustering, and statistical testing. It is validated across diverse benchmarks (MATH, WildChat10K, DS-1000), and its utility is demonstrated through weakness-guided data collection. An interactive demo is also provided to help practitioners explore and interpret the capability tree.
The paper makes a compelling case for the importance of interpretable, capability-based diagnostics of LM capabilities and weaknesses. The tree nodes can be somewhat verbose, but this verbosity enables granular and specific insights into model behavior. Some sections could benefit from more methodological detail (e.g., clustering sensitivity, embedding choices), potentially at the expense of moving secondary applications (e.g., Chatbot Arena critique) into the appendix.
The paper focuses heavily on benchmarking EvalTree against other weakness profiling methods (TEXTDIFF, QUALEVAL), but could benefit from a stronger framing around its primary strength as a diagnostic framework.

**Questions To Authors:**

- How do embedding choices affect tree structure and profile output? Please provide sensitivity analyses or ablations.
- Can EvalTree be adapted for multi-capability instances? This would improve coverage for realistic tasks found in dialogue or multiturn evaluations. Perhaps task decomposition could help here.
- Do you plan to release code for reproducibility and further adoption?
- It would be useful to validate the capability tree against known labels, for example, comparing nodes up to the second layer with MMLU subcategories. There seems to be a placeholder for this in the demo.
- For comparing models more directly, it could be helpful to extend the tree visualization to include per-node rankings over a model set.

Notes:
- Please clarify that the output shown in Figure 2 refers to the ground-truth solution used in the performance metric.
- The paper does not cite related literature on LLM-as-a-judge methods, which is relevant given the prompting-based design used for capability analysis.
- TEXTDIFF and QUALEVAL, while used as baselines, are not mentioned in the Related Work section and should be cited there.
- The Elo rating system, used in part of the analysis, should cite Arpad Elo.
- Consider mentioning TreeEval to avoid confusion with similarly named evaluation methods.

**Reasons To Accept:**

- The paper introduces a strong diagnostic framework for weakness profiling, separating it from traditional leaderboard-style metrics and grounding it in interpretable analysis.
- EvalTree-guided training improves LM accuracy more than generic or category-based data collection, demonstrating real downstream and practical value.
- The method offers better performance at a lower inference cost than TEXTDIFF and QUALEVAL, especially as profile size grows.
- The interactive demo allows users to explore the capability tree for provided examples and inspect node-level performance. This is a valuable contribution that can improve practical adoption.
- As LM evaluation evolves beyond static benchmarks, EvalTree provides a more interpretable, flexible, and fine-grained alternative, making it relevant for both academic and applied NLP settings.

**Reasons To Reject:**

- Capability labels and embeddings are LM-generated, which may reduce consistency if the models used for generating node descriptions and embeddings are not fixed across platforms. Limitations similar to LLM-as-a-judge methods, such as sensitivity to prompt phrasing and model bias, apply here as well.
- The robustness of tree construction is not evaluated. There is no analysis of how the tree structure or node descriptions vary across different embedding models or prompting templates. Sensitivity or ablation studies would strengthen the method's reliability.
- The method currently analyzes all instances, including correctly answered ones. Under compute or budget constraints, it may be more efficient to focus only on failed instances for profiling.
- The baselines (TEXTDIFF and QUALEVAL) from section 4 could be moved into Related Work to streamline the presentation and free space for more detailed methodological explanation.
- The abstract could more clearly reflect the structure of the method: first, capability tree construction; then, weakness profile extraction and comparison with other approaches.

---

> ### Author Response · Authors · 2025-05-31
> **Rebuttal from Authors [1/n]**
>
> We sincerely thank Reviewer SQGa for their thoughtful and constructive review. We appreciate the recognition of EvalTree’s novelty, effectiveness, and efficiency, its strengths as an interpretable diagnostic framework for language model evaluation, its practical value in guiding data collection, and the contribution of the interactive demo to real-world applicability.
>
> Below, we address Reviewer SQGa’s concerns and suggestions in detail.
>
> + Reviewer SQGa raises a point regarding the consistency and reproducibility of EvalTree, as some pipeline components rely on LLM-generated outputs that can vary even with fixed inputs.
>   + We agree this is a reasonable concern. In our experiments, we used GPT-4o-mini and a temperature of 0.0 consistently across all components to ensure internal consistency; however, we acknowledge that this may lead to variability when people want to reproduce our experiments by themselves. To address this, **EvalTree is model-agnostic by design, so its components can be replaced with open-weight LLMs to reduce variance** and improve reproducibility.
>   + To this end, we have conducted new experiments using open-weight LLMs within the EvalTree pipeline. Please see our overall response for further details.
> + Reviewer SQGa asks about how variations in capability annotations and capability embeddings might affect the resulting capability trees.
>   + We agree this is a thoughtful concern given EvalTree’s reliance on LLM-generated annotations and embeddings. To study this, we conducted new ablation experiments. Please see our overall response for further details.
> + Reviewer SQGa suggests that running EvalTree only on failed instances may be more efficient under compute or budget constraints.
>   + We agree that EvalTree can be applied to any subset of instances with a common known prior property, for example, the subset of instances unsolved or solved by an LLM. The capability tree generated from such a subset can indeed provide rich insights into those instances. In practical deployments where resources are constrained, using EvalTree on a filtered subset (e.g., only failures) should be a useful strategy.
>   + However, for the research problem of weakness profiling, we intentionally include all instances for theoretical rigor. We identify weak capabilities by assessing the proportion of errors under a capability rather than the absolute count of errors. This ratio-based consideration is important: suppose a capability contains a disproportionately large number of instances, and the model answers most of them correctly; if we analyzed only the failed instances, this capability could appear to be a major weakness simply because of the volume of examples it contributes, while in fact the model performs well on it; by considering all instances, we avoid such statistical biases for more principled identification of real weaknesses.
> + Reviewer SQGa suggests validating the capability tree against known categorizations, such as comparing second-layer nodes with MMLU subcategories.
>   + We appreciate this suggestion and agree that grounding EvalTree’s structure against a known taxonomy can provide useful insights. To this end, we conducted a new experiment using the MMLU benchmark, which provides 57 human-defined subcategories (e.g., abstract algebra, high school psychology, etc). We compared this human-defined partitioning with a partitioning of instances based on EvalTree; specifically, we focused on the second-layer nodes of the tree, which yielded 46 capability clusters, and each instance was assigned to the second-layer node it descends from. We then computed the [Adjusted Mutual Information (AMI)](https://en.wikipedia.org/wiki/Adjusted_mutual_information) between the EvalTree and MMLU partitionings, obtaining a score of 0.503.
>   + An AMI of 0.503 suggests a moderate-to-strong alignment between the EvalTree-generated partitioning and the MMLU-defined partitioning. This validates that EvalTree captures meaningful, structured distinctions that correspond in part to the topic-based groupings defined by humans.
>   + Furthermore, EvalTree nodes often reflect more than just topical alignment, for instance, goals and reasoning styles, which go beyond what MMLU categories capture; this is why the alignment is not perfect (i.e., close to 1.0). For example, one single capability is described as “*analyzing and synthesizing complex biological, biochemical, and physiological systems and processes to evaluate health impacts and functional interrelationships*”, which contains instances from multiple MMLU categories such as “nutrition,” “clinical_knowledge,” and “college_medicine.”

---

> > ### Author Response · Authors · 2025-05-31
> > **Rebuttal from Authors [1/n, n=2]**
> >
> > + Reviewer SQGa asks about the possibility that EvalTree can be adapted for multi-capability instances, which could be especially relevant for tasks involving multi-turn interactions (e.g., multi-turn dialogue).
> >   + Currently, EvalTree assigns each instance to a single path down the tree, i.e., one child per node, based on its most prominent capability. This simplification is practically sufficient for single-turn or well-scoped tasks, where instances tend to target one primary capability.
> >   + That said, we see clear value in generalizing EvalTree to handle instances that involve multi-turn interactions. One possible design, aligned with the reviewer’s suggestion, is to decompose each instance into multiple capability annotations, compute a separate embedding for each, and allow the instance to be assigned to multiple K-Means clusters during tree construction. This would naturally support hierarchical structures where an instance can appear under multiple branches, better reflecting the compositional nature of tasks involving multi-turn interactions.
> >   + More broadly, we believe there is room for improving capability tree construction, and we outline several such directions in Section 7 (Future Work). We thank Reviewer SQGa for this thoughtful suggestion on potential future direction, and we will incorporate this point in the next revision of Section 7.
> > + Reviewer SQGa asks for showing per-node rankings across a model set in our interactive demo.
> >   + We agree that per-node comparisons are important for comparative evaluation. In fact, the current version of the EvalTree demo already supports this functionality.
> >   + Specifically, users can **right-click on any node** in the demo to access a context menu that includes per-node model rankings. We appreciate the opportunity to clarify this!
> > + Reviewer SQGa makes several helpful suggestions regarding the clarity and organization of the paper writing. These include streamlining the presentation, clarifying some components in the figures, restructuring the abstract, and adding some missing citations. We appreciate these constructive comments. While COLM does not permit changes to the paper during the rebuttal phase, we will incorporate all of these suggestions in our next revision to improve the clarity and completeness of our paper. We thank the reviewer for their careful reading and thoughtful suggestions.

---

> > > ### Comment · Reviewer_SQGa · 2025-06-08
> > >
> > > Thank you for the thorough response. I appreciate the new experiments with open models, the ablations on embeddings and annotations, and the clarifications provided. The authors addressed my concerns well, and I have increased the score from 8 to 9 accordingly.

---

> > > > ### Author Response · Authors · 2025-06-08
> > > >
> > > > Thanks again for your time and thoughtful, encouraging feedback. Your suggestions are constructive and helpful!

---

### Author Response · Authors · 2025-05-31
**Overall Response from Authors**

## Overall Response

We sincerely thank all reviewers for their time and thoughtful feedback! We especially appreciate the shared high recognition across reviews of our work’s novelty, practical value, and strong experimental validation, as well as the enthusiasm for its potential real-world impact.

We would like to highlight that our contributions are two-fold: (i) we are the first to **formulate the research problem** of weakness profiling with a suite of concrete assessments for quantitative method comparison; and (ii) we **propose a weakness profiling method, EvalTree**, for this problem. Together, these two contributions form the foundation of our work. We are encouraged by the reviewers’ appreciation of EvalTree’s design and empirical results, and we are also optimistic that **our formulation of the weakness profiling problem will help set the stage for broader engagement of the community and future development of methods** in this important area.

As noted by Reviewer SQGa and Reviewer DECS, we will certainly release all code and data in the public version of the paper upon acceptance. The code includes implementations of EvalTree and the baseline methods, as well as our assessment code for comparing any weakness profiling methods, to help facilitate future research in this area. In the meantime, we have made an [anonymized GitHub repository](https://github.com/EvalTree-demo/EvalTree-demo.github.io), and an [anonymized interactive demo](https://evaltree-demo.github.io/) of the capability trees built by EvalTree.

We are very grateful for all reviewers’ constructive suggestions on robustness, ablation analysis, and broader positioning. We have addressed all of them in detail in our individual responses and believe they will help us strengthen the clarity and rigor of our work.

## New Experiments

We conducted new ablation experiments to study how variations in capability annotations (from Stage 1) and capability embeddings (from Stage 2) might affect the resulting capability trees.

+ In Stage 1, we adopted a new prompt for capability annotation, and we also separately replaced GPT-4o-mini with two alternatives: GPT-4o and LLaMA-3.3-70B-Instruct (note that the latter is an open-weight model, aligning with our commitment to reproducibility in our response to Reviewer SGQa), respectively; in Stage 2, we replaced OpenAI’s text-embedding-3-small with GritLM-7B [1] embeddings, another open-weight alternative. These four substitutions resulted in four new capability trees on the MATH benchmark, which we have made available for public inspection at our [anonymized interface demo](https://evaltree-demo.github.io/); this brings the total to five trees (including the original one) on MATH.
+ We observe that while the five resulting trees differ in structure and node descriptions, reflecting the inherent subjectivity in LLM-generated annotations, all trees remained reasonable and interpretable upon manual inspection. We believe this variability is expected, as there is no single “ground-truth” capability tree - capability boundaries in real-world tasks are inherently fuzzy.
+ We also compared the weakness profiles generated from different capability trees and found that they exhibit strong qualitative alignment, often identifying similar weaknesses with varying phrasing. For example, in the case of LLaMA-3.1-8B-Instruct on MATH, the weakness profile from the original capability tree has a weakness as: “*Applying Vieta's formulas and manipulating polynomial and trigonometric expressions to analyze relationships, find integer solutions, and determine minimum values*”; in contrast, the capability tree generated using an alternative prompt for capability annotation produced a similarly themed weakness: “*Applying Vieta's formulas to analyze polynomial roots and their relationships through casework and geometric progression*”.

### Reference

[1] [Generative Representational Instruction Tuning](https://arxiv.org/abs/2402.09906)

---

### Decision · Program_Chairs · 2025-07-08

**Decision:**

Accept

**Comment:**

The paper proposes to investigate a model's errors and suggest what is needed to be improved in it in a granular manner, termed "weakness profiling". In addition, the paper proposes a method to perform such profiling.
The paper was uniformly acclaimed by all reviewers in a manner that is rare.
While reviewers do suggest improvements, those stand for me as a sign that nothing is ever perfect, and I do not see a reason to reject the paper.